# Conformational plasticity of a BiP–GRP94 chaperone complex

Joel Cyrille Brenner[1], Linda Charlotte Zirden [2], Lana Buzuk[1], Yasser Almeida-Hernandez [3], Lea Radzuweit[1], Joao Diamantino[1], Farnusch Kaschani [4], Markus Kaiser [4], Elsa Sanchez-Garcia[3], Simon Poepsel [2,5] & Doris Hellerschmied [1]✉

Hsp70 and Hsp90 chaperones and their regulatory cochaperones are critical for maintaining protein homeostasis. Glucose-regulated protein 94 (GRP94), the sole Hsp90 chaperone in the secretory pathway of mammalian cells, is essential for the maturation of important secretory and transmembrane proteins. Without the requirement of cochaperones, the Hsp70 protein BiP controls regulatory conformational changes of GRP94, the structural basis of which has remained elusive. Here we biochemically and structurally characterize the formation of a BiP–GRP94 chaperone complex and its transition to a conformation expected to support the loading of substrate proteins from BiP onto GRP94. BiP initially binds to the open GRP94 dimer through an interaction interface that is conserved among Hsp70 and Hsp90 paralogs. Subsequently, binding of a second BiP protein stabilizes a semiclosed GRP94 dimer, thereby advancing the chaperone cycle. Our findings highlight a fundamental mechanism of direct Hsp70–Hsp90 cooperation, independent of cochaperones.

Cellular protein homeostasis depends on the coordinated activity of molecular chaperones[1]. To this end, the Hsp70 and Hsp90 chaperone families are critical to maintain the integrity of the cellular proteome[2–4]. They are highly conserved across all domains of life, with eukaryotic cells having evolved compartment-specific Hsp70 and Hsp90 paralogs. About 30% of the proteome in eukaryotic cells is processed in the secretory pathway, where the highly abundant BiP and glucose-regulated protein 94 (GRP94) chaperones serve as the exclusive Hsp70–Hsp90 chaperone system[5–8]. BiP and GRP94 safeguard protein homeostasis in the endoplasmic reticulum (ER)[9–11], in post-ER compartments[12,13] and in the extracellular space[14–17]. BiP engages with exposed hydrophobic residues of newly synthesized or misfolded proteins and accordingly serves a broad substrate spectrum[18]. GRP94 is a more specialized chaperone, critical for the quality control of a subset of secretory and transmembrane proteins, including key growth factors and immune

signaling proteins[19]. Previous work on Hsp70 and Hsp90 proteins from different organisms and different organelles established a role for Hsp70 BiP proteins in early substrate folding steps and assigned Hsp90 GRP94 proteins a downstream role in substrate maturation and activation[20–23]. A direct interaction between Hsp70 and Hsp90 proteins is critical for substrate protein handover[24–27]. Mutations impacting conserved Hsp70–Hsp90 interface residues lead to impaired substrate protein folding and growth defects in yeast[24,26,28,29]. To understand protein homeostasis in the secretory pathway, a detailed molecular understanding of the collaboration between GRP94 and BiP is critical.

GRP94, like other Hsp90 proteins, consists of an N-terminal ATPase domain (NTD), a middle domain (MD) and a C-terminal dimerization domain (CTD), all connected by flexible linkers[30] (Fig. 1a). Hsp90 proteins work as dimers, constitutively dimerized by their CTDs[2]. Additional dimerization of the NTDs is subject to regulation, where dynamic

[1]Department of Mechanistic Cell Biology, Center of Medical Biotechnology, Faculty of Biology, University of Duisburg-Essen, Essen, Germany. [2]Center for Molecular Medicine Cologne (CMMC), Faculty of Medicine and University Hospital, University of Cologne, Cologne, Germany. [3]Chair of Computational Bioengineering, Faculty of Biochemical and Chemical Engineering, Technical University of Dortmund, Dortmund, Germany. [4]Department of Chemical Biology, Center for Medical Biotechnology, Faculty of Biology, University of Duisburg-Essen, Essen, Germany. [5]Cologne Excellence Cluster for Cellular Stress Responses in Ageing-Associated Diseases (CECAD), University of Cologne, Cologne, Germany. ✉e-mail: Doris.Hellerschmied@uni-due.de

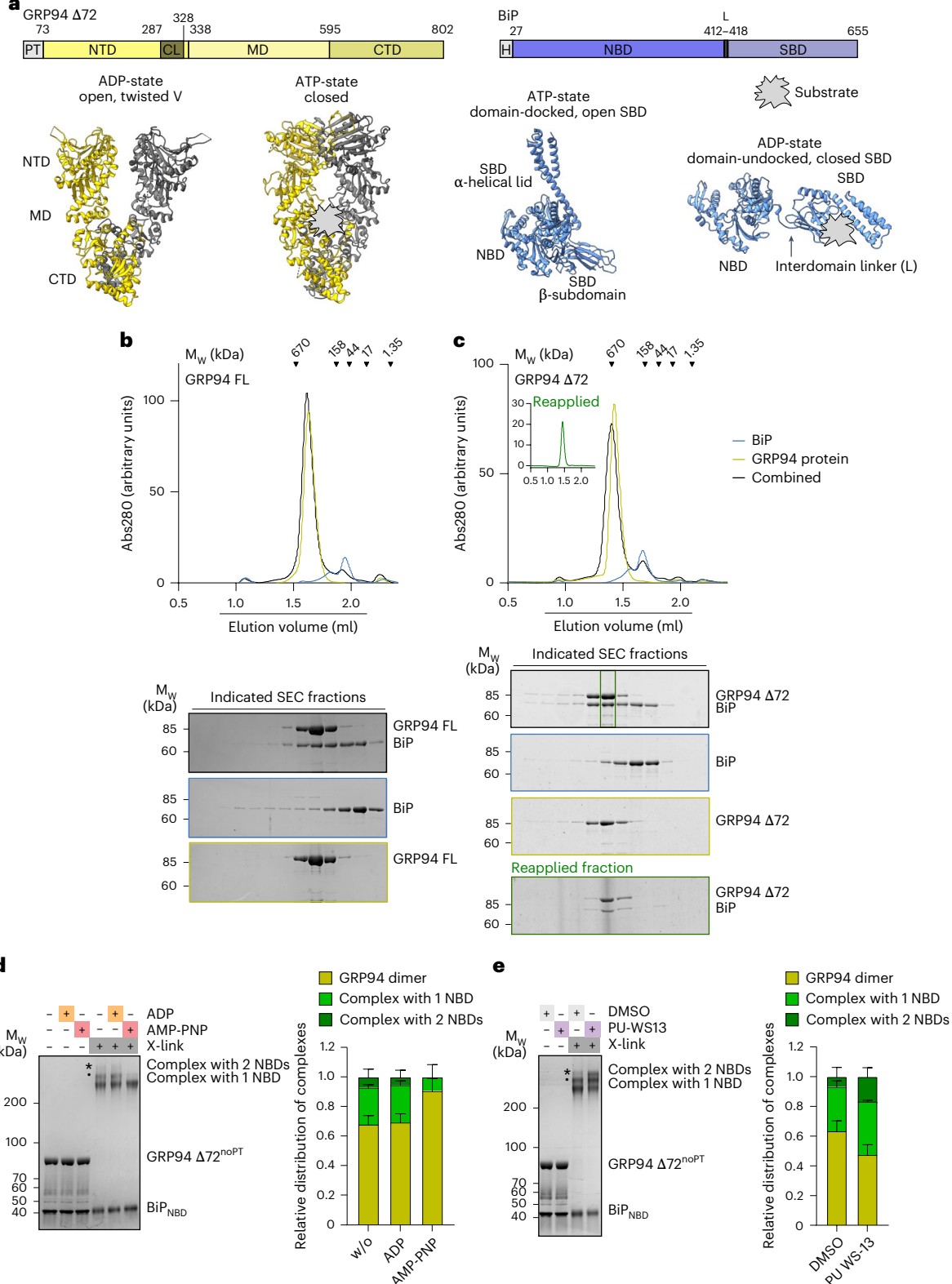

**Fig. 1 | Biochemical characterization of a BiP–GRP94 complex. a**, Crystal structures of the open GRP94 dimer (PDB 2O1V), the closed GRP94 dimer (PDB 5ULS) and domain-docked BiP (PDB 5E84) and a model of BiP based on the domain-undocked conformation of its homolog DnaK (PDB 2KHO). CL, charged linker; L, linker; H, His₆ tag. **b**, Analytical SEC and SDS–PAGE analysis of complex formation between GRP94 FL and BiP. **c**, Same analysis as in **b** for GRP94 Δ72 and BiP. The fraction containing a BiP–GRP94 Δ72 complex (green box) was reapplied to the SEC column. **d**, Left: chemical crosslinking of GRP94 Δ72$^{noPT}$ and BiP$_{NBD}$ in the presence of ADP or AMP-PNP. Right: quantification of the three different species indicated on the gel (*n* = 3 independent replicates; mean + s.d.). **e**, Left: chemical crosslinking of GRP94 Δ72$^{noPT}$ and BiP$_{NBD}$ in the presence of PU-WS13. Right: quantification of the three different species indicated on the gel (*n* = 3 independent replicates; mean + s.d.). Protein bands on SDS–PAGE gels were visualized by Coomassie staining.

opening and closing through intermediate states drive substrate protein processing[2]. Crystal structures of GRP94 show the chaperone in two extreme conformations—an open twisted V[31] and a fully closed conformation[32] (Fig. 1a). Unique features of GRP94 within the Hsp90 family include its N-terminal ER-targeting sequence (1–21) and the pre-N domain (22–72), required for substrate protein maturation[32] and important for the regulation of dimer closure and ATPase activity[31,32], as well as an insertion into its ATP lid[33]. A truncated GRP94 protein, lacking the signal sequence and pre-N domain (GRP94 Δ72), shows increased ATPase activity compared to full-length (FL) GRP94, because of accelerated ATP-driven dimer closure[32]. Cytosolic Hsp90 proteins rely on cochaperones to promote the conformational cycle of opening and closing of the NTDs[2]. For GRP94, only two cochaperones have been described, with a role in conferring substrate selectivity[34,35]. Notably, previous work has demonstrated that BiP acts as a closure-accelerating partner chaperone of GRP94 (refs. 36,37). BiP is an Hsp70-type chaperone with an N-terminal nucleotide-binding domain (NBD) and a C-terminal substrate-binding domain (SBD), which is further subdivided into a β-sheet rich base (SBDβ) and an α-helical lid (SBDα)[4,5] (Fig. 1a). An interdomain linker between $BiP_{NBD}$ and $BiP_{SBD}$ provides the basis for allosteric regulation. ATP binding and hydrolysis of the $BiP_{NBD}$ regulate the conformation of the $BiP_{SBD}$, with the ATP-bound open and the ADP-bound closed conformations representing the most extreme states (Fig. 1a)[38–41]. The SBD-open state is characterized by a high on–off rate for substrate proteins and an interaction interface between $BiP_{NBD}$ and $BiP_{SBD}$[38–41]. It is, therefore, also referred to as the domain-docked conformation. Conversely, the domain-undocked conformation is stabilized by substrate protein bound to the closed $BiP_{SBD}$, which is entirely detached from the $BiP_{NBD}$[39]. In its collaboration with GRP94, BiP serves two major roles that are closely connected; BiP delivers substrates to GRP94 and accelerates GRP94 dimer closure[36,37,42]. Together, these activities should facilitate substrate loading onto GRP94. A recent high-resolution cryo-electron microscopy (cryo-EM) structure provided important insights into substrate transfer between the cytosolic Hsp70–Hsp90 chaperone systems. In the so-called loading complex, resulting from interactions involving Hsp70, Hsp90, the cochaperone Hop and a substrate protein, Hsp90 adopts a semiclosed conformation, where the NTDs have dimerized but are not yet in an ATPase-competent state[24]. Importantly, Hop organizes the loading complex by bridging Hsp70 and Hsp90, stabilizing the semiclosed Hsp90 conformation and extending the substrate interaction interface of the complex[24].

Despite the high conservation and functional similarities between Hsp70–Hsp90 systems across species and organelles, their regulation clearly differs[2,19,43]. Most prominently, bacteria and organellar compartments lack a bridging cochaperone such as Hop. How BiP is structurally integrated into the GRP94 conformational cycle, in the absence of bridging cochaperones, has remained elusive. Here, we describe two distinct states of the BiP–GRP94 complex. In the 'preloading complex', a direct interaction between BiP and the open GRP94 dimer is established through a highly conserved docking site. GRP94 and BiP subsequently move on to a loading complex conformation previously reported for the cytosolic Hsp70–Hsp90–Hop system[24] even in the absence of any cochaperone organizing their interaction. Our biochemical and structural data provide insights into a conserved mechanism of chaperone complex formation and priming for substrate transfer from Hsp70 BiP to Hsp90 GRP94 machineries.

## Results

### A stable complex between BiP and GRP94
To biochemically and structurally characterize the formation of a BiP–GRP94 complex, we reconstituted the interaction between recombinant Strep-tagged GRP94 (residues 23–802) and His₆-tagged BiP (residues 27–655) in vitro. Analytical size-exclusion chromatography (SEC) studies showed that BiP elutes in higher-molecular-weight fractions together with the FL GRP94 wild-type protein (Fig. 1b). We then addressed the

interaction of BiP and the GRP94-D149N nucleotide-binding mutant, expected to preferentially populate an open dimer conformation, and the GRP94-E103A ATP-hydrolysis mutant, expected to reside mainly in a closed state[32,44]. Compared to GRP94 wild-type protein, we did not observe major differences in the coelution of BiP with GRP94-D149N or GRP94-E103A (Extended Data Fig. 1a). We also tested the interaction of BiP with GRP94 Δ72, which lacks the pre-N domain (Fig. 1a), and we observed a stable complex, illustrated by the fact that, upon reapplying the complex to the SEC column, BiP and GRP94 Δ72 still coeluted in a single peak (Fig. 1c). By chemical crosslinking, we could further stabilize this complex (Extended Data Fig. 1b) and test the effect of nucleotides on complex formation. These experiments were performed with GRP94 $\Delta72^{noPT}$, where the purification tag (PT) was removed by protease cleavage. In the presence of the nonhydrolyzable ATP analog, AMP-PNP, we saw a trend of reduced BiP–GRP94 complex formation (Extended Data Fig. 2a). In this setup, AMP-PNP affects the conformation of both chaperones, BiP and GRP94. BiP shifts to the SBD-open, domain-docked conformation and GRP94 shifts to the closed conformation (Fig. 1a). To specifically assess the effect of the GRP94 closed state on complex formation, we used a BiP truncation construct comprising only its NBD ($BiP_{NBD}$). The $BiP_{NBD}$, irrespective of its nucleotide state, retains GRP94 binding[36,37]. In analytical SEC experiments, $BiP_{NBD}$ did not coelute with GRP94 Δ72 (Extended Data Fig. 2b). However, in chemical crosslinking experiments, we could visualize a complex between the $BiP_{NBD}$ and GRP94 $\Delta72^{noPT}$. We presume that the bands for this complex are more distinct than with FL BiP because the latter oligomerizes in solution (Fig. 1d and Extended Data Fig. 2a). The addition of ADP did not affect $BiP_{NBD}$–GRP94 $\Delta72^{noPT}$ complex formation, whereas an excess of AMP-PNP, which shifts the GRP94 dimer to the closed conformation[32], strongly reduced the interaction (Fig. 1d). The nucleotide dependence of the interaction was recapitulated for GRP94 FL and the GRP94 Δ72 construct (Extended Data Fig. 2c). We then also tested the effect of PU-WS13, a GRP94 ATPase inhibitor that is expected to prevent full closure of the $GRP94_{NTD}$[17,45]. Using GRP94 FL or GRP94 $\Delta72^{noPT}$ and FL BiP protein, we observed a potential positive effect of the GRP94 inhibitor on chaperone complex formation (Extended Data Fig. 2d). Because crosslinking with $BiP_{NBD}$ yields more distinct bands, we used this construct to assess the effect of PU-WS13, showing that PU-WS13 increases the fraction of GRP94 in complex with $BiP_{NBD}$ (Fig. 1e). Taken together, we characterized the interaction between BiP and GRP94, which is reduced by AMP-PNP and favored by PU-WS13. These data suggest that the complete closure of the GRP94 dimer counteracts binding to BiP.

### Assessing the topology of a BiP–GRP94 complex by crosslinking mass spectrometry
To gain initial structural insight into a BiP–GRP94 complex, we performed crosslinking mass spectrometry (XL-MS) experiments. We used disuccinimidyldibutyric urea (DSBU), a homobifunctional crosslinker reactive to primary amines with a spacer length of 12.5 Å. DSBU primarily attaches to lysines but can also react with serines, threonines or tyrosines. We reconstituted complexes between BiP and GRP94 FL or GRP94 Δ72. Upon crosslinking with DSBU, SDS–PAGE gel bands containing high-molecular-weight protein complexes were analyzed by MS (Fig. 2a and Supplementary Tables 1 and 2). In both datasets, the crosslinks between $BiP_{NBD}$ and GRP94 suggest that the two chaperones adopt a similar conformation to their cytosolic paralogs, Hsp70 and Hsp90, in the previously described loading complex[24]. Figure 2b shows a homology model of BiP and GRP94 based on the cytosolic loading complex (Protein Data Bank (PDB) 7KW7), with BiP in the domain-undocked conformation and GRP94 in a semiclosed state. The majority of $BiP_{NBD}$–GRP94 crosslinks map to two interaction interfaces (I and II) (Fig. 2bc, Extended Data Fig. 3a and Supplementary Tables 1 and 2), matching interactions described for the cytosolic loading complex. The crosslinks outlining interface I link the $GRP94_{MD}$ helix 455–476 to $BiP_{NBD}$ lobe IIA. The crosslinks that indicate the formation

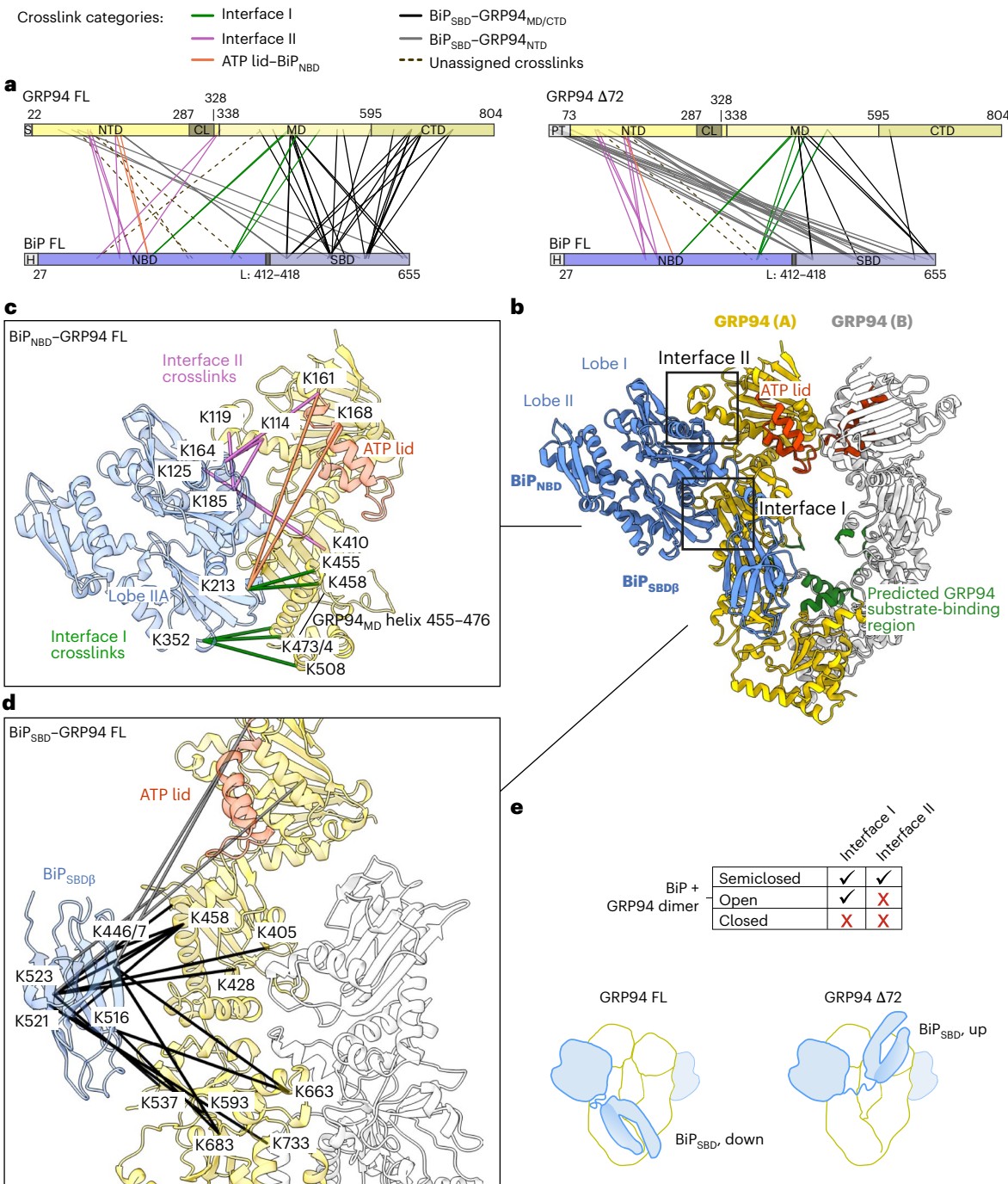

**Fig. 2 | XL-MS outlines the topology of a BiP−GRP94 complex. a**, Map of crosslinks identified in the BiP−GRP94 FL (left) and BiP−GRP94 Δ72 (right) samples. **b**, Homology model of BiP−GRP94 on the basis of the cytosolic loading complex (PDB 7KW7). The ATP lid is indicated in red. Residues indicated as the predicted GRP94 substrate-binding region (dark green) include the amphipathic helix N657−Q668 as identified in the cytosolic Hsp90 paralog and additional residues F398−Y401, K428, I497, E498 and Y575 as identified in Huck et al.[32]

**c**, Crosslinks outlining interface I, interface II and the ATP lid−BiP$_{NBD}$ interactions from the BiP−GRP94 FL XL-MS dataset mapped onto the homology model shown in **b**. BiP$_{SBD}$ is omitted for clarity. **d**, Crosslinks from the BiP$_{SBD}$ to GRP94 mapped onto the homology model shown in **b**. BiP$_{NBD}$ is omitted for clarity. **e**, Summary of XL-MS results. S, Strep tag. Crosslinks to GRP94 protomer A are shown, which represent the closest distances for all residues shown in **c,d** except for K733.

of interface II link the central GRP94$_{NTD}$ helix (residues K114 and K119) and the ATP lid (residues K161 and K168) to the BiP$_{NBD}$ lobe I. Additional crosslinks between BiP residue K213 and GRP94 residues K161 and K168 suggest that the GRP94 ATP lid may be slightly closer to the BiP$_{NBD}$ than shown in the homology model (Fig. 3c and Extended Data Fig. 3a, coral crosslinks). Crosslinks outlining interfaces I and II were also prevalent in XL-MS datasets of BiP and BiP$_{NBD}$ in complex with GRP94 Δ72$^{noPT}$ in

the presence of PU-WS13 (Extended Data Fig. 3b and Supplementary Tables 3 and 4).

BiP binding to GRP94 is compatible with the open and semiclosed GRP94 conformations, allowing for interactions within interface I or interfaces I and II, respectively (Extended Data Fig. 3c). In the GRP94 closed conformation, the ATP lid would clash with residues of BiP$_{NBD}$ lobe IA, preventing an efficient interaction (Extended Data Fig. 3c).

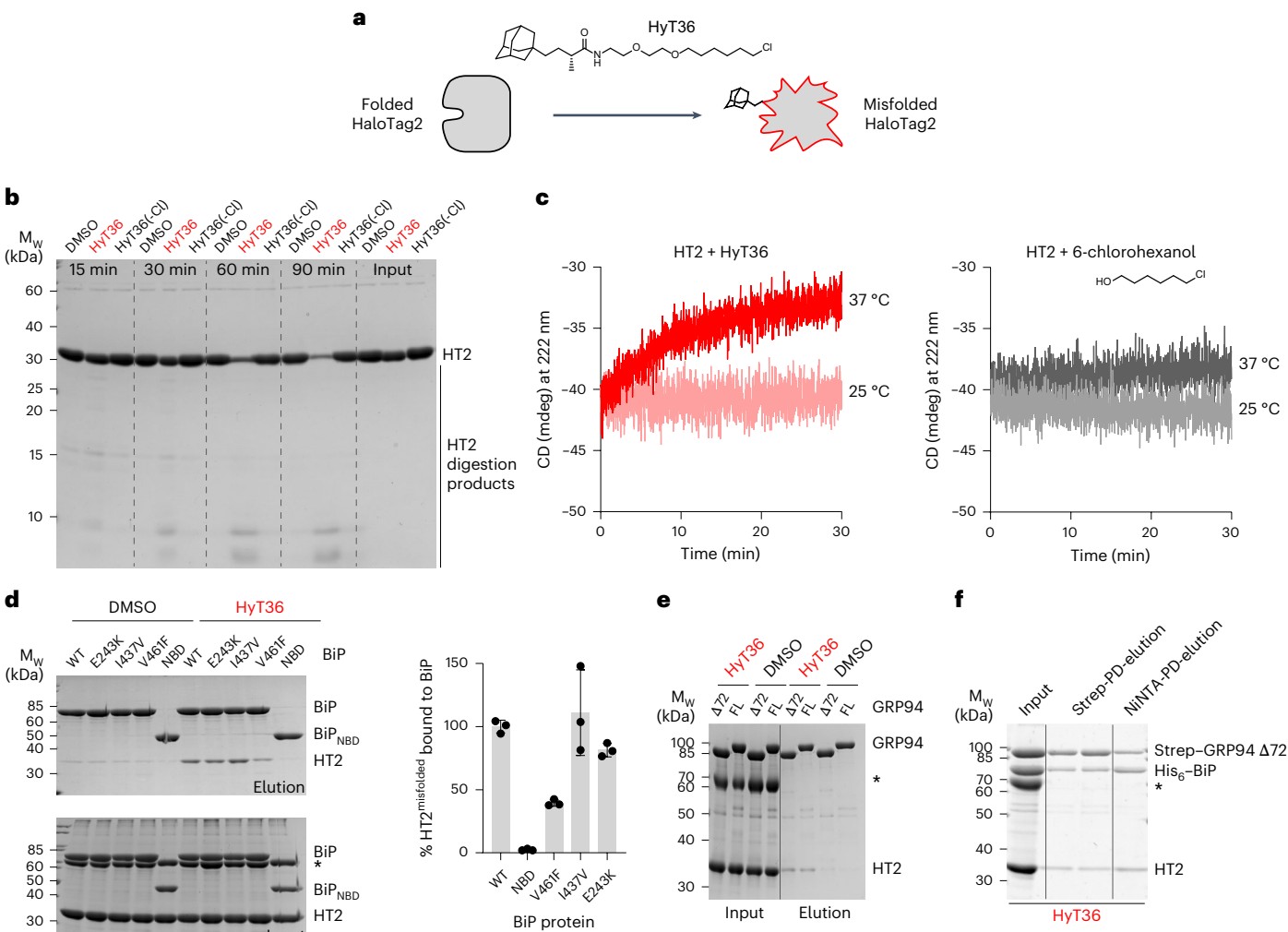

**Fig. 3 | Hydrophobic tagging to establish a BiP–GRP94–substrate complex.**
**a**, Cartoon model of HT2 misfolding upon covalent conjugation to HyT36.
**b**, Limited proteolysis with trypsin to assess the folding state of HT2 conjugated with HyT36, incubated with the HyT36(-Cl) control or DMSO. **c**, CD spectroscopy of HT2 conjugated to HyT36 or 6-chlorohexanol. Kinetic measurements at the indicated temperatures are shown. **d**, Pulldown experiments to study the interaction of His₆–

BiP with HT2^misfolded–HyT36 compared to DMSO control. For quantification, data were normalized to the wild-type BiP elution sample ($n = 3$ independent replicates; mean ± s.d.). **e**, Same analysis as in **d** for GRP94 FL and GRP94 Δ72. **f**, Sequential pulldown of Strep–GRP94 Δ72 and His₆–BiP to isolate a GRP94 Δ72–BiP–HT2^misfolded complex. The asterisks in **d**–**f** indicate BSA in the interaction buffer. Protein bands on SDS–PAGE gels were visualized by Coomassie staining.

This effect is consistent with our biochemical data showing that the AMP-PNP-bound and, thus, closed GRP94 dimer shows a reduced interaction with the BiP_NBD (Fig. 1d). To characterize the conformation of GRP94, we analyzed intra-GRP94 crosslinks. We measured distances between crosslinked residues within one GRP94 protomer and across the GRP94 dimer in the open and the semiclosed GRP94 conformations (Supplementary Tables 1–4). The GRP94_MD and GRP94_CTD residues exhibit minimal conformational changes between the open and semiclosed states (root-mean-square deviation (r.m.s.d.) of 2.7 Å). Crosslinks within these regions agree with published structures, validating the quality of the crosslinking data (Supplementary Tables 1–4). As these crosslinks cannot distinguish between different GRP94 conformations, we focused on crosslinks between the GRP94_NTD and GRP94_MD. In the open GRP94 conformation, the central GRP94_NTD helix (residue K114) and the GRP94_NTD ATP lid (residues K161 and K168) are in close proximity to the GRP94_MD helix 455–476, compatible with crosslinks within a single protomer and across the GRP94 dimer between these GRP94_NTD and GRP94_MD regions (Extended Data Fig. 4a, purple crosslinks). Upon rotation of the GRP94_NTD to the semiclosed conformation, the central GRP94_NTD helix and the ATP lid move away from the GRP94_MD. At the same time, helix H1 is repositioned to participate

in the GRP94_NTD dimer interface. This repositioning now supports a crosslink between K95 residues across the GRP94 dimer. Additionally, K95 and K97 link to residues of the GRP94_MD helix 455–476, within a single protomer and across protomers (Extended Data Fig. 4a, orange crosslinks). Crosslinks supporting either of the two GRP94 conformations were found in all XL-MS datasets (Supplementary Tables 1–4), suggesting that our samples contain open (twisted V) and semiclosed GRP94 dimers.

Crosslinks within the BiP_SBD confirm that, when in complex with GRP94, it resides in the closed conformation (Extended Data Fig. 4b and Supplementary Tables 1, 2 and 4), which is typically the substrate-bound state. The closed BiP_SBD is fully dissociated from the BiP_NBD and free to move around the interdomain linker[39,46]. Notably, the position of the BiP_SBD with respect to GRP94 differs between GRP94 FL and GRP94 Δ72. The majority of BiP_SBD–GRP94 Δ72 crosslinks place the BiP_SBD in the proximity of the GRP94_NTD and suggest an interaction with the N-terminal GRP94 Δ72 PT (Fig. 2a and Supplementary Table 2). The BiP–GRP94 FL XL-MS dataset positions the BiP_SBD in close proximity of the GRP94_MD and GRP94_CTD (Fig. 2d and Supplementary Table 1). Notably, on the basis of previous work, the lumen of the GRP94_MD and GRP94_CTD dimer contains the predicted substrate-binding region

(Fig. 2b)[24,32,47]. Moreover, this BiP$_{SBD}$ position matches the conformation of the Hsp70–SBDβ resolved in the cytosolic loading complex[24], which contained a substrate protein (Supplementary Table 1). Taken together, our data suggest that the BiP$_{SBD}$ extends toward the predicted substrate-binding region of GRP94, even in the absence of substrate (Fig. 2e).

## Hydrophobic tagging to form chaperone–substrate complexes

To better understand how GRP94 and a BiP–GRP94 complex interact with substrate proteins, we established interaction studies with a model substrate. The HaloTag 2 (HT2) domain, a self-labeling protein tag[48], can covalently conjugate to a hydrophobic tag, which induces HT2 misfolding in cells—a process called hydrophobic tagging (Fig. 3a)[49–52]. In mammalian cells, BiP and GRP94 interact with misfolded HT2 (HT2$^{misfolded}$)-based model substrates targeted to the Golgi apparatus and the ER[53,54] (Extended Data Fig. 5).

We characterized the HyT36-induced misfolding process of HT2 in vitro by limited proteolysis using trypsin. Upon incubation of HT2 with HyT36, the protein is more susceptible to tryptic digest, indicating a misfolded state (Fig. 3b). HT2 misfolding is dependent on the covalent conjugation to HyT36 because, in the presence of a control compound, HyT36(-Cl), the protein remains stable (Fig. 3b). In circular dichroism (CD) spectroscopy experiments, we analyzed the kinetics of misfolding (Fig. 3c). The α-helical content of HT2 (measured by an increase in signal at 222 nm) reached its minimum at ~30-min incubation at 37 °C (as opposed to 25 °C), revealing experimental conditions that can be applied in chaperone–substrate interaction studies. Conjugation of HT2 to 6-chlorohexanol, an HT ligand without any functional group, did not affect the stability of the HT2 protein at 25 °C or 37 °C (Fig. 3c).

The interaction of Hsp70-type chaperones with misfolded substrate proteins is well characterized[55–57]. For in vitro experiments, substrate proteins are often destabilized with nonspecific chemical denaturing agents. Upon dilution of the denaturing agent, chaperones can then be added to study chaperone–substrate interactions[58,59]. With the in vitro hydrophobic tagging system, HT2 misfolding can be induced directly in the presence of the Hsp70 BiP, providing access to HT2$^{misfolded}$ states as they arise. We conjugated HyT36 to HT2 by simple mixing of the two components, followed by the addition of BiP and ATP and finally incubation at 37 °C to induce HT2 misfolding. Under these conditions, HT2$^{misfolded}$ is directly bound by BiP, as observed in pulldown assays (DMSO versus HyT36; Fig. 3d). Deletion of the BiP SBD entirely abolished the BiP–HT2$^{misfolded}$ interaction (Fig. 3d, BiP$_{NBD}$). A previously described substrate-binding mutant, BiP-V461F (refs. 60–62), showed reduced interaction with HT2$^{misfolded}$. We also tested BiP-I437V, a substitution described to stabilize the closed (domain-undocked) conformation[39], and BiP-E243K, a substitution at the BiP$_{NBD}$–BiP$_{SBD}$ interface, described to have reduced interaction with GRP94 (ref. 37). While BiP-I437V did not consistently affect the binding of the substrate, BiP-E243K showed a trend toward slightly reduced substrate binding (Fig. 3d). We then performed similar experiments with GRP94 FL and GRP94 Δ72, which also bound to HT2 upon misfolding (Fig. 3e). Lastly, we performed a sequential pulldown experiment. By first capturing Strep-tagged GRP94 Δ72 and then His$_6$-tagged BiP, we could isolate a GRP94 Δ72–BiP–HT2$^{misfolded}$ complex (Fig. 3f). In XL-MS experiments including the HT2-based substrate, we unfortunately did not detect crosslinks to HT2, preventing a detailed positioning of the substrate protein within the chaperone complex.

## BiP binds to the open GRP94 dimer in the preloading complex

To study the topology and conformations of a BiP–GRP94 Δ72 complex, we performed single-particle negative-stain EM. Even though we could not detect any crosslinks to HT2, we hypothesized that the misfolded substrate may stabilize or support the formation of the BiP–GRP94 chaperone complex; therefore, we included HT2$^{misfolded}$ in our sample preparation. To establish a BiP–GRP94 Δ72–HT2$^{misfolded}$ complex, we

first incubated BiP with HT2 in the presence of HyT36 and ATP and then added GRP94 Δ72. To enrich the multichaperone–substrate complex, we performed gradient fixation (GraFix)[63], which resulted in the formation of a GRP94 dimer (complex 0) and two complexes composed of a GRP94 dimer, HT2 and one or two BiP molecules, respectively (complexes I and II) (Extended Data Fig. 6a). GraFix fraction 12 contained the largest amounts of complex I and II with little amount of free GRP94 dimer and was, therefore, used for negative-stain EM (Extended Data Fig. 6a,b). As expected, reference-free two-dimensional (2D) classifications and ab initio three-dimensional (3D) reconstructions indicated sample heterogeneity (Extended Data Figs. 6c and 7). Nevertheless, the ab initio 3D reconstructions in comparison to known GRP94 structures showed two predominant complexes corresponding to a GRP94 dimer with one (complex I) and two (complex II) BiP proteins bound (Extended Data Fig. 7).

In the complex I reconstruction, we identified a density corresponding to a GRP94 dimer with a gap between the NTDs (Fig. 4a). The additional density attached to GRP94 protomer A resembles a BiP$_{NBD}$ domain with a characteristic globular shape outlining lobes I and II and an indentation corresponding to the nucleotide-binding cleft (Fig. 4a). Accordingly, we chose an open twisted V GRP94 Δ72 structure (PDB 2O1V)[31] and the isolated BiP$_{NBD}$ (PDB 6DWS) as the starting structures for molecular modeling. After an initial rigid-body fit, we performed a map-fitting refinement of the complex with Rosetta[64,65]. Both the total_score and the elect_dens_fast values of the obtained models negatively correlated with the real-space correlation coefficient (RSCC) values (Extended Data Fig. 8a), probably because the models could fit most of the density (Fig. 4b). We presume that the BiP$_{SBD}$ is rather flexible and, therefore, largely not visible in the EM map. A density close to the GRP94$_{NTD}$ protomer B, not accounted for in our model, may correspond to the BiP interdomain linker and/or parts of the BiP$_{SBD}$ (Fig. 4ab). We refer to this state of the BiP–GRP94 complex as the preloading complex, a conformation that exists before dimerization of the GRP94$_{NTD}$ (that is, before the full engagement of GRP94 with substrate protein). As expected, for a BiP–GRP94 complex with GRP94 in the open twisted V conformation, the preloading complex model is supported by our crosslinking data outlining interface I, whereas interface II has not formed yet (Fig. 2b and Extended Data Fig. 8b). The preloading complex depicts the initial engagement of BiP with the open GRP94 dimer, the substrate-loading competent form.

## BiP bound to a semiclosed GRP94 conformation

In addition to the preloading complex, we obtained a reconstruction that indicated the binding of two BiP molecules to a GRP94 dimer (complex II) and largely resembled the cytosolic loading complex reported by Wang et al.[24]. Our ab initio reconstruction shows that the GRP94$_{NTD}$ established a contact at the center of the dimer, indicating a semiclosed conformation (Fig. 4c). Furthermore, the NBDs of the two outlined BiP proteins established interactions with the GRP94$_{MD}$ (interface I) and GRP94$_{NTD}$ (interface II). To initiate molecular modeling, we obtained a homology model of GRP94 based on the loading complex (PDB 7KW7)[24]. For BiP1, we used the isolated NBD (PDB 6DWS) and, for BiP2, we included the BiP$_{SBD}$ β-subdomain using a homology model of a domain-undocked conformation based on the structure of its *Escherichia coli* homolog DnaK (PDB 2KHO)[66]. The individual models were placed into the EM reconstruction and structural refinement allowed to fit the extended conformation of BiP2$_{NBD–SBD}$ into the density (Fig. 4d). We refer to this state of the BiP–GRP94 complex as the loading complex, given the similarity to the previously reported Hsp70–Hsp90 structure (Extended Data Fig. 8c)[24]. The fitting scores and the RSCC for the loading complex as compared to the preloading complex lacked a clear correlation (Extended Data Fig. 8a). This could be explained by the larger size and complexity of the loading complex model and by the fact that, at a lower contour level, the EM reconstruction showed a density connected to BiP1$_{NBD}$ that is not accommodated by the final

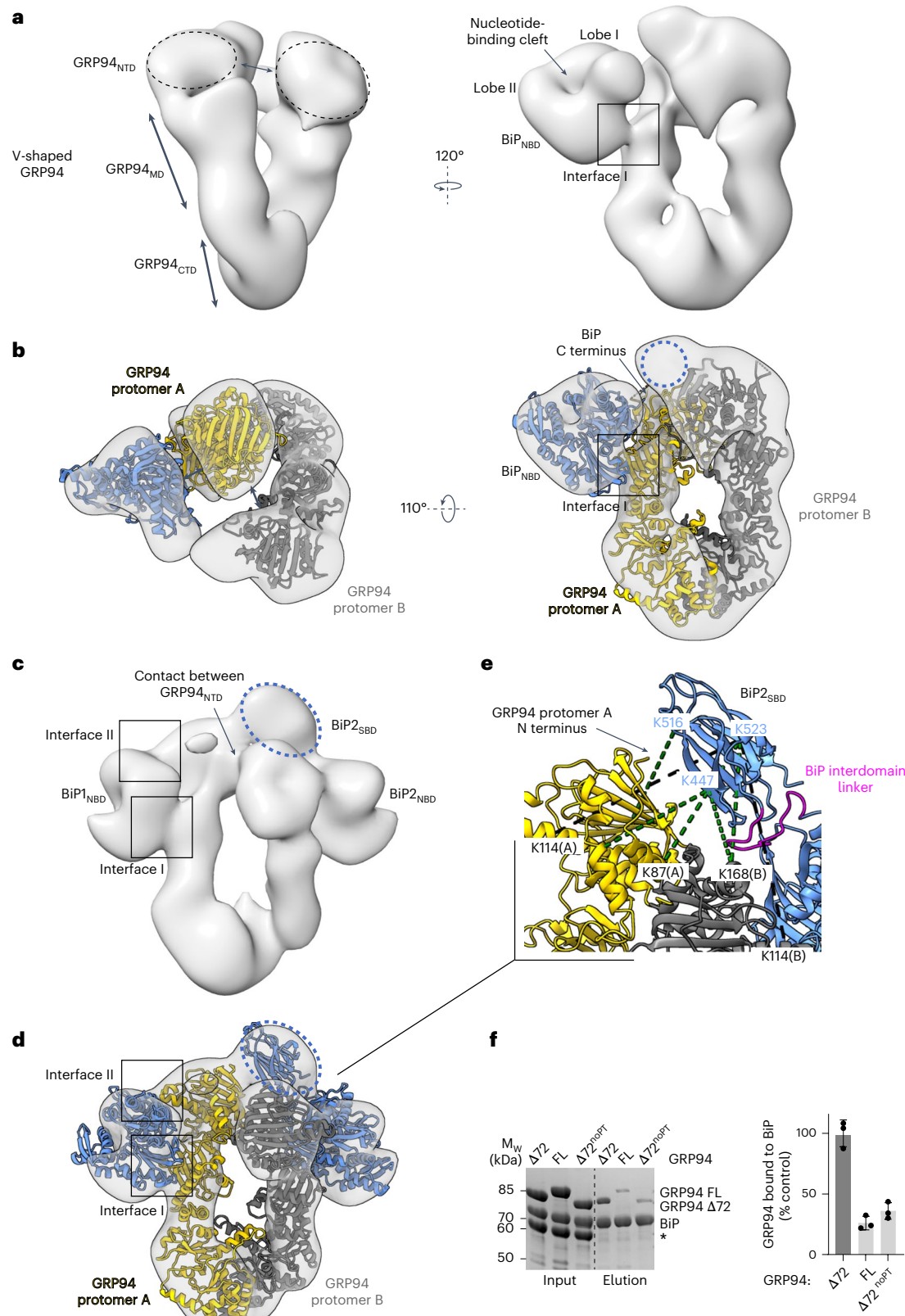

**Fig. 4 | Structure of the BiP–GRP94 complex in two different conformations.** **a**,**b**, Negative-stain EM reconstruction (**a**) and molecular model (**b**) of the BiP–GRP94 preloading complex. **c**,**d**, Negative-stain EM reconstruction (**c**) and molecular model (**d**) of the BiP–GRP94 loading complex. The dotted blue circles indicate the potential position of the BiP$_{SBD}$. **e**, Crosslinks identified between BiP$_{SBD}$ and GRP94 Δ72 mapped onto the loading complex structure. Green lines indicate crosslinks < 35 Å; black lines indicate crosslinks > 35 Å. The arrow

indicates the N terminus of GRP94 that is the attachment point of the PT. **f**, Left: Ni-NTA pulldown analysis of His$_6$–BiP with GRP94 Δ72, GRP94 FL and GRP94 Δ72$^{noPT}$ (exact sequences of the N termini in Extended Data Fig. 9a). The asterisk indicates BSA in the interaction buffer. Right: quantification of the pulldown experiment (n = 3 independent replicates; mean ± s.d.). Protein bands on SDS–PAGE gels were visualized by Coomassie staining.

model (Extended Data Fig. 8a). We did not observe any additional densities that could be assigned to the HT2$^{misfolded}$, which may be because of its flexibility and size or resolution limits of negative-stain EM.

In both EM reconstructions, the BiP$_{SBD}$ appears to extend toward the NTD of the opposing GRP94 protomer (Fig. 4a–d). In the preloading complex, residual density in combination with the BiP$_{NBD}$ C terminus indicates the position of the BiP$_{SBD}$ (Fig. 4b). In the loading complex, the defined BiP2$_{SBD}$ extends toward the GRP94$_{NTD}$ protomer A and an additional interaction between the BiP interdomain linker and the GRP94$_{NTD}$ of protomer B is formed (Fig. 4d,e). This position of the BiP2$_{SBD}$ with respect to GRP94 and its PT is in agreement with our crosslinking data (Fig. 4e). Upon close inspection of the PT sequence, we identified a hydrophobic stretch, which may serve as a BiP substrate mimetic (Extended Data Fig. 9a). To address the contribution of the GRP94 PT to the interaction, we removed it by protease cleavage and performed comparative pulldown analyses (Fig. 4f). Removal of the tag reduced GRP94 binding to BiP by ~65%, suggesting that the tag contributes to the stability of the chaperone complex. In line with this, we also observed ~75% less GRP94 FL protein pulled down with BiP compared to GRP94 Δ72 (Fig. 4f). The FL protein, while carrying a Strep tag, does not contain the predicted BiP substrate sequence (Extended Data Fig. 9a).

The binding of the BiP$_{SBD}$ to the GRP94 Δ72 PT contributes an additional interaction interface and we presume that it prohibits transition to the fully closed GRP94 conformation, which is expected to induce complex disassembly (Fig. 1d). Taken together, stabilization of the BiP–GRP94 Δ72 complex by this inadvertently engineered interaction allowed us to capture two chaperone complex transition states.

## BiP$_{NBD}$–GRP94 interactions in preloading and loading complex

In the two obtained EM reconstructions, the preferred position of the BiP$_{SBD}$ is determined by the GRP94 Δ72 PT. Given that, in the domain-undocked state, the BiP$_{NBD}$ acts independently of the BiP$_{SBD}$, we further characterized BiP$_{NBD}$–GRP94 interactions (Fig. 5a). In the preloading complex, the BiP$_{NBD}$ and GRP94 showed one major contact point with an interface area (GRP94$_{MD}$:BiP$_{NBD}$) of 1,072.5 Å$^2$ (Fig. 5b). According to our model, the long GRP94$_{MD}$ helix (residues 455–476) packs against BiP$_{NBD}$ lobe IIA, thereby forming electrostatic interactions involving BiP residues D238, N239 and E243 and GRP94 residues K462, K463, R466 and K467 (Fig. 5b). Additionally, the tip of a GRP94$_{MD}$ β-sheet inserts between BiP$_{NBD}$ lobes IA and IIA. This interaction matches interface I in the previously described cytosolic loading complex[24] (Fig. 5b). Mutations impacting interface I impair the interaction and functional cooperation between Hsp70 and Hsp90 chaperones[25,26,28,37]. Consistently, BiP-E243K (E218 in cytosolic Hsp70) and GRP94-K463E or GRP94-K467E (K414 and K419 in cytosolic Hsp90) strongly reduced BiP–GRP94 Δ72 complex formation in analytical SEC studies and pulldown experiments (Fig. 5c,d). For all three mutant proteins, an ~85% reduction in the amount of GRP94 Δ72 copurified with BiP was observed in pulldown experiments (Fig. 5c).

Our model of complex II matches the overall topology of the cytosolic loading complex, with slightly modified interaction interfaces between the BiP and GRP94 proteins (Extended Data Fig. 8c and 9b). At the heart of the complex is the semiclosed GRP94, supported by the rotation of the GRP94$_{NTD}$ compared to the preloading complex and by the binding of two BiP molecules (Supplementary Video 1). The resulting interface II from the GRP94$_{NTD}$–BiP1 interaction indicates conserved hydrophobic contacts with the C-terminal part of the GRP94$_{NTD}$ helix (residues 117–121) and polar contacts with R116 (Fig. 5e). To evaluate the importance of the conserved interface II, we analyzed the GRP94-R116G mutant (R60G in cytosolic Hsp90 and R46G in yeast Hsp90), previously shown to decrease the interaction between cytosolic Hsp70 and Hsp90 proteins and to reduce the fitness of yeast cells[24]. Pulldown experiments showed an ~30% decrease in the ability of this mutant to bind BiP (Fig. 5f), corroborating the importance of the conserved interface II

for the stability of a BiP–GRP94 Δ72 complex. The comparatively small reduction in the bound GRP94 Δ72 mutant protein can be attributed to the fact that interface I is critical for preloading and loading complex conformations, while the role of interface II is limited to stabilization of the loading complex (Fig. 4b). Overall, our data suggest that, through interactions along interface I, BiP establishes the initial contact with GRP94, which is still in an open conformation. Single substitutions at interface I strongly impair the interaction between BiP and GRP94. The binding of a second BiP molecule through interface I and the subsequent formation of interface II stabilize the GRP94 semiclosed state.

## Discussion

GRP94 is a structurally dynamic molecular chaperone critical for the folding, maturation and degradation of important secretory and transmembrane proteins[19]. In the apo or ADP-bound state, the GRP94 dimer resides in an open conformation[31,32,36]. To tightly interact with its substrate proteins, the dimer closes[32]. Cytosolic Hsp90 proteins rely on cochaperones for controlled ATP-driven transitioning from the open to the closed state[2,30]. In the secretory pathway, the Hsp70 protein BiP regulates GRP94 function[36,37,42]. Extensive characterization of Hsp90 proteins shows that closure intermediate states have a critical role in the chaperone cycle[24,67–69] and that BiP is critical for the stabilization of such GRP94 states[36].

Building on this previous work, our structural and biochemical data provide insight into the BiP–GRP94 preloading and loading complex conformations (Fig. 6). These transition states likely became visible by an interaction of BiP with the PT of GRP94 Δ72 acting as a BiP substrate mimetic and preventing the full closure of the GRP94$_{NTD}$. In the context of the stabilized complex, we could characterize interactions between BiP$_{NBD}$ and GRP94. On the basis of our findings, we propose a model according to which the stepwise binding of BiP controls GRP94 dimer closure. Initially, one BiP molecule binds to the open twisted V conformation of GRP94 by establishing interactions at interface I. The twisted V conformation indicates the GRP94 apo or ADP-bound state. Furthermore, BiP needs to reside in the ADP-bound, extended conformation to interact with GRP94 (ref. 37), which is consistent with our XL-MS data. The residues within the BiP$_{NBD}$ and GRP94$_{MD}$ critical to this interaction site are highly conserved from bacteria to the different Hsp70 and Hsp90 chaperone systems in mammalian cells[24–26,28,37], indicating that the formation of the preloading complex may also be conserved. Following this initial contact, additional interactions between BiP and GRP94 are formed along interface II, favoring the loading complex conformation. Importantly, in the loading complex, a second BiP molecule is bound to GRP94, increasing the interface area of the complex. We propose that the simultaneous engagement of both GRP94 protomers in interface II promotes GRP94$_{NTD}$ rotation (Fig. 6). Upon rotation, the GRP94$_{NTD}$ is poised to dimerize and assume the semiclosed conformation. The interactions along interface II thereby define the semiclosed GRP94 state and may underlie the GRP94 closure-accelerating activity of BiP[36,37]. We presume that, under physiological conditions, GRP94 is bound to ATP in the loading complex to enable the transition to the maturation complex. Interestingly, our data suggest that binding of the GRP94 ATPase inhibitor PU-WS13 (ref. 17) affects the BiP–GRP94 complex. According to the crystal structure of GRP94$_{NTD}$ in complex with the closely related PU-H36 (PDB 8TF0), inhibitor binding rearranges the GRP94$_{NTD}$ ATP lid (residues 161–197) and the GRP94 N-terminal residues (including helix H1) compared to the apo state but does not support full closure of the GRP94$_{NTD}$[17,45]. We hypothesize that PU-WS13 stabilizes a GRP94 semiclosed state, which promotes the interaction with BiP.

The cytosolic loading complex is stabilized by the cochaperone Hop through direct contacts with all complex components—Hsp70, Hsp90 and the substrate protein. The semiclosed Hsp90 conformation is defined by direct Hop–Hsp90 interactions and by the binding of two Hsp70 proteins[24]. Even in the absence of a bridging cochaperone such

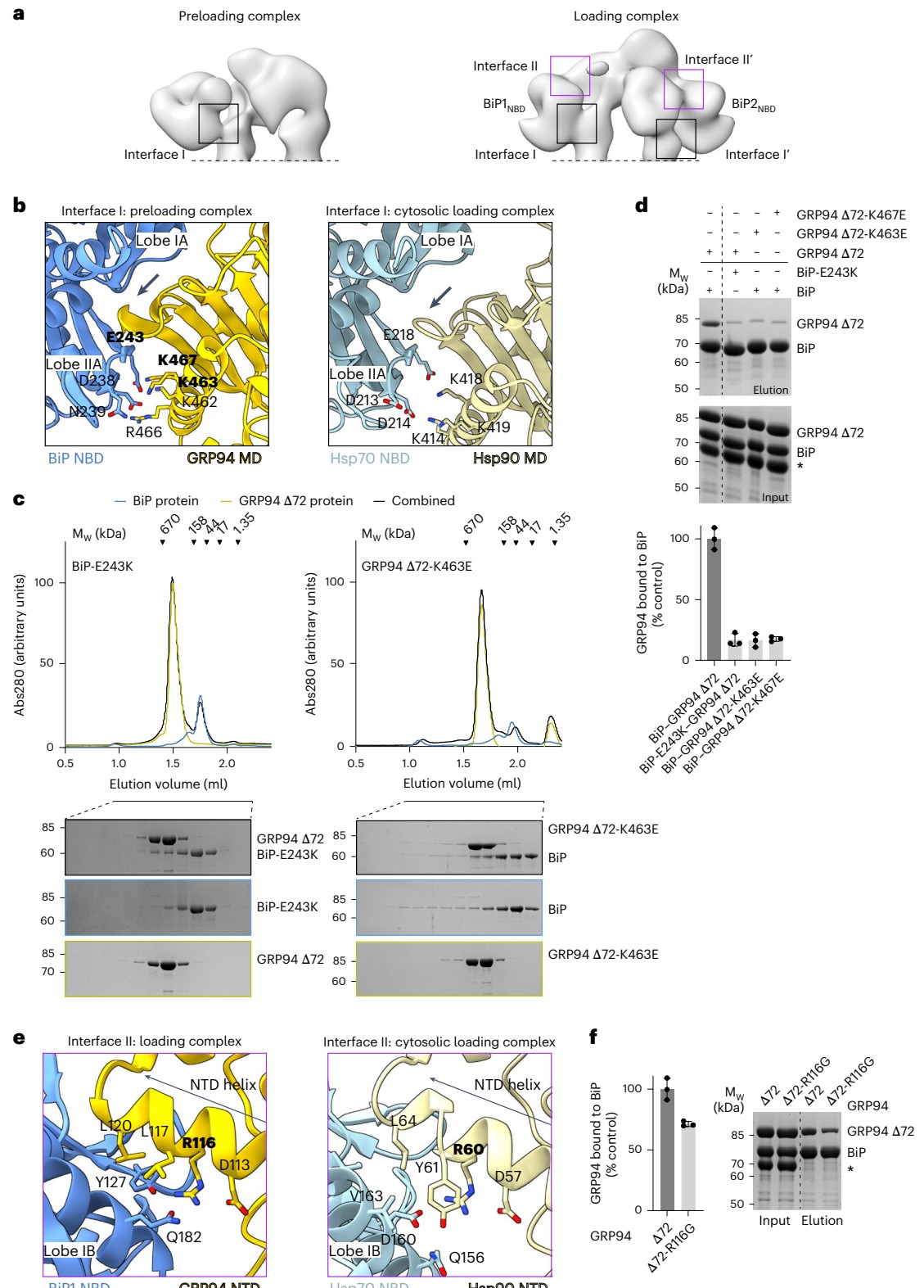

**Fig. 5 | Interactions between BiP$_{NBD}$ and GRP94. a**, Overview of interfaces I and II in the preloading and loading complex conformations. **b**, Zoomed-in view of interface I of the BiP–GRP94 preloading complex. Interface I of the cytosolic (Hsp70–Hsp90) loading complex is shown for comparison. The arrow points to the GRP94$_{MD}$ β-sheet loop inserted between BiP$_{NBD}$ lobes IA and IIA. Residues in bold were used for mutational analysis. **c**, Analytical SEC and SDS–PAGE analysis of complex formation between BiP and GRP94 Δ72 interface I mutants. **d**, Ni-NTA pulldown analysis of His$_6$-tagged BiP and Strep-tagged GRP94 Δ72 interface I

mutants and quantification of the pulldown experiment ($n$ = 3 independent replicates; mean ± s.d.). The asterisk indicates BSA in the interaction buffer. **e**, Zoomed-in view of interface II of the BiP–GRP94 loading complex. Interface II of the cytosolic loading complex is shown for comparison. **f**, Ni-NTA pulldown analysis of His$_6$-tagged BiP and Strep-tagged GRP94 Δ72-R116G interface II mutant and quantification of the pulldown experiment ($n$ = 3 independent replicates; mean ± sd). The asterisk indicates BSA in the interaction buffer. Protein bands on SDS–PAGE gels were visualized by Coomassie staining.

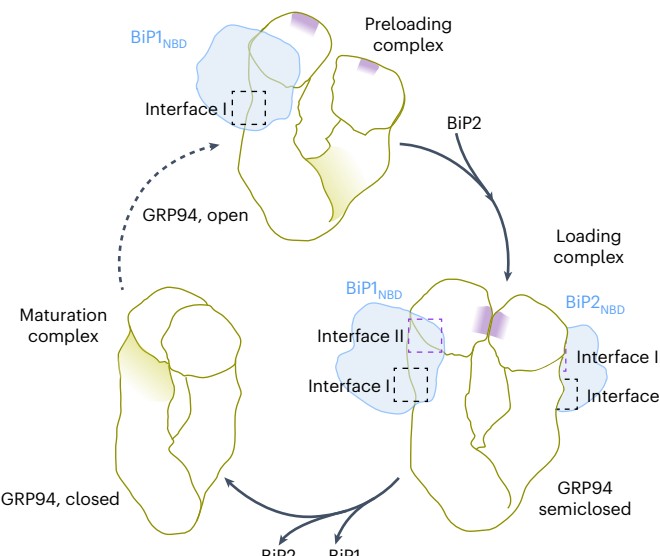

**Fig. 6 | Model of the conformational plasticity of the BiP–GRP94 complex along the GRP94 chaperone cycle.** In the preloading complex, one BiP molecule is bound to an open GRP94 dimer, which is in the apo or ADP-bound state. The binding of a second BiP molecule supports transitioning into the loading complex, where GRP94 is in a semiclosed conformation. ATP binding to GRP94 may support the partial closure of the ATP lid and GRP94$_{NTD}$ dimerization under physiological conditions. A similar effect may be produced by PU-WS13. Both BiP molecules bound to GRP94 reside in the ADP-bound, domain-undocked conformation. Finally, the GRP94 dimer transitions into the fully closed state and BiP concurrently dissociates. We presume that ATP binding to BiP drives this process. The relative position of the GRP94$_{NTD}$ semiclosed dimer interface is indicated in purple.

as Hop, two BiP proteins are important for stabilizing the semiclosed GRP94 conformation. The additional role of Hop in substrate binding may not be required for secretory and transmembrane protein folding or may be substituted by still uncharacterized cochaperones. Given the similarities between the BiP–GRP94 and cytosolic Hsp70–Hsp90 complexes and the conservation of interface residues, we believe that the stepwise BiP (Hsp70) binding model to drive GRP94 (Hsp90) dimer closure (Fig. 6) may also apply to more sophisticated systems, where cochaperones have evolved to support and fine-tune this principal mechanism of Hsp70–Hsp90 cooperation. Consistent with previous work in the Hsp90 field[24], our data show that the ATP-bound, fully closed GRP94 does not interact with BiP (Fig. 1d). Closing of the GRP94$_{NTD}$ ATP lid breaks contacts at interface II and subsequent swapping of the GRP94$_{NTD}$ and ATP binding to BiP leaves substrate-bound GRP94 and BiP in the open, domain-docked conformation primed for another round of substrate recruitment (Fig. 6). Agard and colleagues also proposed a stepwise release of Hsp70 proteins during the transition from the loading complex to the so-called maturation complex, which may equally apply to the simpler BiP–GRP94 system[24,70] (Fig. 6).

The coordinated interactions between BiP and GRP94 are aimed at transferring substrates from BiP$_{SBD}$ to GRP94. We applied the chemical biology tool hydrophobic tagging to translate chaperone–substrate interactions previously identified in cells[53,54] (Extended Data Fig. 5) into an in vitro reconstituted system. This shows that hydrophobic tagging provides an efficient approach for in vitro reconstitution of chaperone–substrate complexes, as demonstrated by our usage for two different types of chaperones, BiP and GRP94 (Fig. 3). Inducing HT2 misfolding in the presence of chaperones may provide an advantage in establishing chaperone–substrate complexes over an experimental setup where protein misfolding and chaperone-binding reactions are separated. BiP and GRP94 individually interacted with HT2$^{misfolded}$ but could also form a multichaperone–substrate complex. One limitation of the HT2-based

system is its limited applicability in crosslinking experiments with DSBU and glutaraldehyde. Unfortunately, HT2 could not be visualized in our XL-MS or GraFix EM experiments, which may be because of its low lysine content (seven lysines, 2.3% of its sequence). For comparison, lysine residues make up 9% (58 residues) and 9.7% (77 residues) of the sequences of BiP and GRP94, respectively. Furthermore, HT2 may be shielded by the substrate-binding sites of BiP and GRP94, preventing the formation of crosslinks flanking the substrate-binding regions. The use of HT2 fusion proteins may be an avenue in overcoming this limitation in future research. While we could not visualize a substrate-bound chaperone complex by XL-MS or EM, we expect, by extrapolating from the cytosolic loading complex, a state where the substrate protein is engaged by BiP and GRP94 simultaneously. We presume that interactions between the substrate protein and GRP94 drive the formation of the loading complex, in addition to interactions between GRP94$_{NTD}$ and BiP$_{NBD}$, further facilitating substrate transfer. A substrate protein would bridge the two chaperones, providing additional interaction interfaces, and may support the transition to the GRP94 semiclosed state. The two orientations of BiP$_{SBD}$ with respect to BiP$_{NBD}$ captured in our samples underline its flexibility in the domain-undocked state. In-solution studies have previously highlighted that the crystal structures of the domain-docked and domain-undocked BiP conformations (Fig. 1a) represent endpoint states[39,40]. This flexibility of the BiP$_{SBD}$ can facilitate the delivery of a highly diverse range of substrate proteins to GRP94. In the context of the GRP94 FL protein, BiP$_{SBD}$ is in close proximity to GRP94$_{MD}$ and GRP94$_{CTD}$, suggesting that closed BiP$_{SBD}$ may have a preference for the orientation toward the predicted GRP94 substrate-binding region.

Our negative-stain EM and XL-MS data highlight striking similarities between the comparatively simple BiP–GRP94 chaperone system and their intricate Hsp70–Hsp90 paralogs and point to a highly conserved mode of substrate transfer among Hsp70 and Hsp90 systems. Accordingly, Hsp70 (BiP) proteins have a conserved role, not only in the delivery of substrate proteins but also in the activation of Hsp90 (GRP94) proteins. On the basis of the identified BiP–GRP94 preloading and loading complexes, it will be interesting to define ER-specific regulatory mechanisms. Post-translational modifications may influence transitioning between the different states of the GRP94 chaperone cycle. Regulatory modifications may affect the stability and lifetime of the different BiP–GRP94 complex states and could also change the role of the chaperones from collaborating in protein folding to supporting protein degradation.

## Online content

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

## Methods

### Reagents and antibodies

HyT36 and HyT36(-Cl) were kind gifts from C. Crews and G. Burslem[50,51]. 6-Chlorohexanol (Sigma-Aldrich, C45008), trypsin (Sigma-Aldrich, T8003), DSBU (Thermo Fisher, A35459), glutaraldehyde (Thermo Fisher), ATP (Sigma-Aldrich, A6419), AMP-PNP (Sigma-Aldrich, A2647), ADP (ARCOS Organics, 164670010), PU-WS13 (MedChemExpress, HY-18680), HALTS1 (*N*-(3,5-dichloro-2-ethoxybenzyl)-2*H*-tetrazol-5-amine)[71] (ChemBridge Corporation, 9074451), cycloheximide (Sigma-Aldrich, 01810), doxycycline hyclate (Sigma-Aldrich, DD9891), anti-His5 (Santa Cruz Biotechnologies, SC-8036), anti-Strep (Qiagen, 34850), anti-BiP C50B12 (Cell Signaling, 3177), anti-GRP94 (Proteintech, 14700-1-AP), anti-HA tag C29F4 (Cell Signaling 3724), anti-HT (Promega, G9211), goat anti-mouse HRP-coupled secondary antibody (Invitrogen, 31444) and goat anti-rabbit HRP-coupled secondary antibody (Invitrogen, 31460) were from commercial suppliers.

### Plasmids and molecular cloning

The expression plasmids His$_6$–linker–TEV site–GRP94 (73–802) in pET151 and His$_6$–BiP (27–655) in pNIC28-Bsa4 were a kind gift from T. Street[37]. GRP94 was recloned into pET21d to yield an expression plasmid encoding Strep–linker–TEV site–GRP94 (73–802). Details on the N-terminal PT can be found in Extended Data Fig. 9a. Strep–GRP94 FL was cloned from C2C12 complementary DNA into pET21d by Gibson assembly (New England Biolabs Hifi Builder). His$_6$–BiP$_{NBD}$ (27–411) was cloned from His$_6$–BiP in pNIC28-Bsa4 by Gibson assembly into the same vector. pcDNA5/FRT/TO-EGFP-HT2 (ref. 53) was used as a template to clone HT2-His$_6$ in pET21a. Spot–HT2 was cloned from His$_6$–HT2 into pET22b-SUMO. All point mutations were introduced by site-directed mutagenesis. All primers are listed in Supplementary Table 5.

### Protein expression and purification

All proteins were expressed in *Escherichia coli* BL21(DE3) cells. Protein expression was induced with 250 µM IPTG for 16 h at 18 °C for GRP94 and BiP proteins and for 5 h at 25 °C for His$_6$–SUMO–HT2 and HT2–His$_6$. Cells were harvested by centrifugation, resuspended in 50 mM Na$_2$HPO$_4$ pH 8.0 and 300 mM NaCl for His$_6$–SUMO–HT2 and HT2–His$_6$, 100 mM HEPES pH 8.0, 400 mM NaCl, 10 mM imidazole and 0.5 mM tris(carboxyethyl)phosphine (TCEP) for His$_6$–BiP proteins and 100 mM HEPES pH 8.0, 400 mM NaCl and 0.5 mM TCEP for Strep-tagged GRP94 proteins and lysed by sonication.

For His$_6$-tagged proteins, cleared cell lysate was applied to a His-Trap HP column (Cytiva). The column was washed by applying a stepwise imidazole gradient and the His$_6$-tagged protein was eluted in 150 mM imidazole in lysis buffer. His$_6$–SUMO–Spot–HT2 was digested with His$_6$–Ulp1 and subjected to another round of HisTrap affinity chromatography to isolate Spot–HT2. For Strep-tagged proteins, cleared cell lysate was applied to a StrepTrap HP column (Cytiva) and eluted with 2.5 mM D-desthiobiotin in lysis buffer. In the case where the PT was removed by protease digest with His$_6$–GST-TEV-S219V, GRP94 Δ72 was exchanged into 100 mM HEPES pH 8.0, 150 mM NaCl, and 2 mM DTT. As a final step, all proteins were subjected to SEC using a Superdex 200 16/600 column equilibrated with 50 mM HEPES pH 7.5 and 150 mM NaCl for the HT2 proteins, supplemented with 0.5 mM TCEP for BiP and GRP94 proteins. Fractions containing the protein of interest were concentrated, flash-frozen in liquid N$_2$ and stored at −70 °C.

### Hydrophobic tagging

Spot–HT2 or HT2–His$_6$ was incubated with HyT36, HyT36(-Cl) or DMSO for 10 min at room temperature. To induce misfolding, samples were incubated at 37 °C.

### Limited proteolysis

First, 10 µM His$_6$–HT2 reacted with 12 µM HyT36 or in the presence of 12 µM HyT36(-Cl) or DMSO was incubated with 0.1 µM trypsin (Sigma-Aldrich) for 15, 30, 60 and 90 min at 37 °C. Reactions were stopped by the addition of SDS–PAGE sample buffer and resolved on SDS–PAGE gels.

### CD spectroscopy

CD spectra of 10 µM Spot-HT2 reacted with 10 µM HyT36 or 10 µM 6-chlorohexanol or in the presence of DMSO were recorded in 20 mM Tris pH 7.5 and 100 mM NaCl at 25 °C and 37 °C with a Jasco Spectropolarimeter J-710 connected to the temperature control unit Jasco PFD-350s with a Julabo F-250 compact recirculation cooler at 0.2-nm steps with ten scans. Data were analyzed in GraphPad Prism. Given the presence of DMSO in the samples, spectra were analyzed from 218 to 260 nm.

### Sample preparation for EM

To prepare protein complexes for negative-stain EM, 16 µM BiP and 32 µM HT2 reacted with 32 µM HyT36 were incubated for 30 min at 37 °C in 50 mM HEPES pH 7.5, 50 mM NaCl, 2 mM MgCl$_2$ and 50 µM ATP. Then, 16 µM GRP94 was added to the mix and the sample was incubated for another 30 min at 37 °C. Gradients were generated using a range from 5% glycerol in 50 mM HEPES, 50 mM NaCl and 1 mM CaCl$_2$ to 45% glycerol in 50 mM HEPES, 50 mM NaCl, 1 mM CaCl$_2$ and 0.5% glutaraldehyde using the Master Gradient 108 (BioComp Instruments; program: 5–45% sucrose for 45 s). The generated gradient was incubated for 60 min at 4 °C. The protein samples were carefully applied to the gradient and centrifuged for 16 h at 130,000*g* at 4 °C (Optima XPN-80 Ultracentrifuge, Beckman Coulter). Next, 200 µl fractions were taken from the top of the gradient and the glutaraldehyde was quenched with 20 mM Tris pH 7.5. Gradient fractions were analyzed by SDS–PAGE.

To generate negative-stain grids, 4 µl of the GraFix fraction was applied to copper 400-mesh grids with continuous carbon (EM Sciences) after glow discharge, incubated for 30 s, immediately blotted with filter paper and stained by five successive short incubations of 2% (w/v) uranyl formate (Science Services). The excess stain was removed with filter paper and the grids were dried before imaging.

### Collection and processing of negative-stain EM data

Grids were imaged using a Talos L120C microscope (Thermo Fisher) operated at 120 keV with a complementary metal–oxide–semiconductor camera (Ceta 16M). Micrographs were acquired at a nominal pixel size of 1.89 Å and total dose of 25 e$^-$ per Å$^2$. Data were processed in cryoSPARC[72], using CTFFIND4 for CTF estimation[73], blob picker for particle picking and iterative rounds of reference-free 2D classification. Classes representing damaged protein or artifacts were removed in several rounds of 2D classification. Approximately ~434,000 particles corresponding to representative 2D class averages were used for ab initio reconstructions with several classes and heterogeneous refinements (Extended Data Figs. 6b and 7).

### Molecular modeling

The preloading state of the GRP94–BiP complex was modeled with the structures of the 'open' ADP-bound GRP94 (PDB 2O1V)[31] and the structure of BiP$_{NBD}$ (PDB 6DWS) as inputs. The region comprising residues 393–408 of GRP94 was modeled using MODELLER[74,75]. For the loading complex, a homology model of GRP94 was obtained with the SWISS-MODEL web server[76] using the 'semiclosed' structure of Hsp90 (PDB 7KW7)[24] as a template. This complex encompassed one BiP$_{NBD}$ chain as in the preloading complex and a model of the extended conformation of BiP that also included the SBD (residues 30–635). This structure was built using SWISS-MODEL with the *E. coli* Hsp70 homolog DnaK (PDB 2KHO)[66] as a template. For this model, the region of residues 285–330 was truncated similarly to the open structure.

Before fitting into the EM density, the α-helical C-terminal region of the model of the extended structure of BiP was truncated (residues 530–635 were removed). For the modeling based on the obtained negative-stain EM maps, a first rigid-body fitting was performed with

UCSF Chimera[77] on the densities obtained for the preloading and loading complexes. Subsequently, a refinement of the fitting was performed with Rosetta's 'relax' application[64] with the specific flags for cryo-EM refinement[65]. We used the crosslinking information corresponding to NBD–NBD, SBD–SBD and NBD–SBD pairs as constraints for the fitting. For all the fittings, the input resolution of the density maps was 20 Å. For both preloading and loading complexes, a Cartesian fitting was used and 5,000 decoys were generated. For the evaluation, Rosetta's 'ref2015' (refs. [78],[79]) and 'elec_dens_fast'[80,81] score functions were used, the latter with a weight of 10. Additionally, the RSCC was calculated with Rosetta's 'density_tools' application. The interface area was measured with UCSF ChimeraX[82]. UCSF Chimera[82] and PyMol (Molecular Graphics System, version 2.0, Schrödinger) were used for the analysis and visualization of the structures.

## Crosslinking of chaperone complexes for SDS–PAGE analysis and XL-MS

All reactions were carried out in 50 mM HEPES pH 7.5, 50 mM NaCl and 2 mM MgCl$_2$. For generating the BiP–GRP94 complex, 4 μM BiP or BiP$_{NBD}$ and 4 μM GRP94 Δ72, GRP94 Δ72$^{noPT}$ or GRP94 FL were incubated for 60 min at 37 °C. For the reactions containing PU-WS13, either 100 μM PU-WS13 or 1% DMSO was incubated with 4 μM GRP94 Δ72$^{noPT}$ for 30 min at 37 °C and then 4.0 μM BiP and 50 μM ATP were added before incubating for an additional 30 min at 37 °C. Samples were brought to room temperature (10 min) and DSBU (Thermo Fisher, A35459) was added to a final concentration of 0.5 mM. Crosslinking reactions were incubated for 20 min at room temperature and quenched with 20 mM Tris pH 7.5; the protein complexes were isolated by SDS–PAGE.

## In-gel digestion of DSBU-crosslinked proteins

Gel regions containing crosslinked proteins were excised and washed twice with water and twice with 100 mM ammonium bicarbonate solution. The proteins were reduced with 10 mM TCEP (30 min, 62 °C) while gently shaking and subsequently alkylated by adding 55 mM iodoacetamide (30 min, room temperature, in the dark). The gel slabs were subsequently washed three times with acetonitrile (50 μl each). After the last wash, the gel slabs were completely dried by using a vacuum concentrator (Eppendorf) for 5 min. The dried gel pieces were next incubated with 200 μl of 10 ng μl$^{-1}$ trypsin (37 °C, 16 h, vigorous shaking). The digestion was stopped by addition of formic acid to a final concentration of 5% (v/v) formic acid. The digestion supernatant was transferred to a fresh Eppendorf tube. The remaining gel pieces were subsequently washed three times with acetonitrile (50 μl each). The supernatants of these washes were combined with the recovered digestion mix and dried in a vacuum concentrator (Eppendorf).

The cleared tryptic digests were then desalted on homemade C18 stag tips as described previously[83]. Briefly, peptides were immobilized and washed on a two-disc C18 stage tip. After elution from the stage tips, samples were dried using a vacuum concentrator (Eppendorf); the peptides were taken up in 0.1% formic acid solution (10 μl) and directly used for liquid chromatography (LC)–MS/MS experiments, as described below.

## LC–MS/MS

MS experiments were performed on an Orbitrap LUMOS instrument (Thermo Fisher) coupled to an EASY-nLC 1200 ultraperformance LC (UPLC) system (Thermo Fisher). The UPLC was operated in the one-column mode. The analytical column was a fused silica capillary (75 μm × 28 cm) with an integrated fritted emitter (CoAnn Technologies) packed in house with Kinetex 1.7-μm core–shell beads (Phenomenex). The analytical column was encased by a column oven (Sonation PRSO-V2) and attached to a nanospray flex ion source (Thermo Fisher). The column oven temperature was set to 50 °C during sample loading and data acquisition. The LC was equipped with two mobile phases: solvent A (0.2% formic acid, 2% acetonitrile and 97.8% H$_2$O) and solvent

B (0.2% formic acid, 80% acetonitrile and 19.8% H$_2$O). All solvents were of UPLC grade (Honeywell). Peptides were directly loaded onto the analytical column with a maximum flow rate that would not exceed the set pressure limit of 980 bar (usually around 0.4–0.6 μl min$^{-1}$). Peptides were subsequently separated on the analytical column by running a 67-min or 105-min gradient of solvent A and solvent B. The MS instrument was controlled by the Orbitrap Fusion Lumos Tune application and operated using Xcalibur software. Detailed LC–MS settings can be found in Supplementary Information.

## XL-MS data analysis

Thermo RAW files were converted to the mzML or mfg file format using ProteoWizard (version 3.0.23018-60066e9)[84] with default settings and submitted to Xisearch (version 1.7.6.7) for further analysis[85]. For each search, a dedicated database containing only the sequences of the investigated proteins was used. The following settings were applied: MS1 error tolerance of 6 ppm, MS2 error tolerance of 20 ppm, trypsin digestion with two missed cleavages allowed, carbamidomethylation on cysteine as the fixed modification, oxidation on methionine as the variable modification, DSBU as the crosslinker (linked amino acids included lysine, serine, threonine and tyrosine), NH$_2$, OH, Bu and BuUr as linked modifications, a maximum of three modifications per peptide and fragment $b$ and $y$ ions. The resulting candidates from Xisearch were filtered to a 5% false discovery rate (FDR) on the residue pair level using the software xiFDR (version 2.1.5.5)[86]. Xisearch also calculates a score (reported in Supplementary Tables 1–4), which considers the goodness of the fit of the spectrum to the peptide pair based on several factors such as fragment mass error and number of (crosslinked) fragments[85]. A penalty is applied for crosslinks to serine, threonine and tyrosine as these are a side reaction of DSBU as compared to lysine and the N terminus of the protein. The mgf files, xiFDR results and FASTA sequences were then imported and visualized by XiView to generate the results (Fig. 2a and Extended Data Fig. 3b)[87]. Distance measurements were performed in PyMol or ChimeraX. The r.m.s.d. values of GRP94 protomers A and B in PDB 2O1V and the model of GRP94 generated on the basis of PDB 7KW7 were 0.8 Å and 0.7 Å respectively. The r.m.s.d. was calculated in PyMol and the flexible GRP94 charged linker region was omitted for the calculation. We, therefore, report only one value for interprotomer distances in the Supplementary Tables 1–4.

## Pulldowns and sequential pulldown

All reactions were carried out in 50 mM HEPES pH 7.5, 50 mM NaCl, 2 mM MgCl$_2$, 50 μM ATP and 3 μM BSA. Then, 8 μM HT2 reacted with 12 μM HyT36 was incubated with 4 μM His$_6$-tagged BiP protein and/or 4 μM Strep-tagged GRP94 protein (as indicated in the individual figures) for 60 min at 37 °C. For pulldowns involving untagged GRP94 Δ72, the PT was removed by incubation with His$_6$–GST–TEV-S219V protease for 2 h at room temperature in 50 mM HEPES pH 7.5, 50 mM NaCl and 1 mM CaCl$_2$. Upon protease cleavage, untagged GRP94 Δ72 protein was isolated from the flowthrough of Streptactin following Ni-NTA purification. Next, 4 μM His$_6$-tagged BiP protein and 4 μM Strep-tagged or untagged GRP94 protein were incubated for 60 min at 37 °C.

For His$_6$-tagged BiP protein pulldown, samples were incubated with Ni-NTA beads (Qiagen) for 60 min at 37 °C, washed three times with 50 mM HEPES pH 7.5, 50 mM NaCl, 10 mM imidazole and 1 mM CaCl$_2$ and finally eluted with 500 mM imidazole in wash buffer. For Strep-tagged GRP94 protein pulldown, samples were applied to StrepTactin 4 flowXT gravity flow columns (IBA), washed three times with 50 mM HEPES pH 7.5, 50 mM NaCl and 1 mM CaCl$_2$ and finally eluted with 50 mM biotin in wash buffer. For the sequential pulldown, the elution of a Strep pulldown described above was used as the input for a Ni-NTA pulldown as described above. All input and elution fractions were applied to SDS–PAGE.

## Analytical SEC

BiP and GRP94 proteins were incubated at a final concentration of 8 µM (unless otherwise indicated) for 60 min at 37 °C and applied to a Superose 6 PC 3.2/300 column in 50 mM HEPES pH 7.5, 50 mM NaCl and 1 mM $CaCl_2$ on an Ettan LC system. Indicated fractions were analyzed by SDS–PAGE.

## Glutaraldehyde crosslinking

For glutaraldehyde crosslinking experiments, 8 µM GRP94 and 8 µM $BiP_{NBD}$ were incubated separately or together in 50 mM HEPES pH 7.5, 50 mM NaCl and 2 mM $MgCl_2$ in the presence or absence of 1 mM AMP-PNP for 60 min at 37 °C. Samples were brought to room temperature (10 min) and 0.006% glutaraldehyde was added. After 5 min at room temperature, reactions were quenched with 20 mM Tris pH 7.5 and the samples were analyzed by SDS–PAGE and western blot using anti-His$_5$ (Santa Cruz Biotechnologies, SC-8036; 1:1,000 dilution) and anti-Strep (Qiagen, 34850; 1:1,000 dilution) antibodies. Bands were detected by chemiluminescence on a BioRad ChemiDoc imager.

## Pulldown from HEK293 cells

Stable HEK293 Flp-in Trex cells (Thermo Fisher, R78007) expressing B4GT–HA–EGFP–HT2 were previously described in Serebrenik et al.[51]. The cells were grown in 15-cm dishes in DMEM (Invitrogen, Life Technologies, 31966047), 10% FBS (Life Technologies, 10500056) and 1% penicillin–streptomycin (Invitrogen, Life technologies, 15140122) in 5% $CO_2$ at 37 °C. The cells were treated with 0.1 µg ml$^{-1}$ doxycycline hyclate for 24 h, followed by cycloheximide treatment at 20 µg ml$^{-1}$ 1 h before 10 µM HyT36 or control (DMSO) treatments for an additional 4 h. HALTS1 treatment at 10 µM was performed simultaneously with the doxycycline hyclate treatment. The cells were then lysed in 10 mM Tris pH 7.5, 150 mM NaCl, 0.5 mM EDTA and 0.5% Triton X-100 (Fisher Scientific, BP151-100) supplemented with cOmplete protease inhibitor cocktail (Roche, 11836145001). The cell lysates were clarified by centrifugation at 17,000g for 20 min at 4 °C. The supernatant was then incubated with GFP-trap agarose beads (ChromoTek) for 1 h at 4 °C. The beads were washed three times with 10 mM Tris pH 7.5, 150 mM NaCl, 0.5 mM EDTA and 0.05% Triton X-100 and bound proteins were eluted by boiling the beads in SDS–PAGE sample buffer for 5 min. Samples were applied to SDS–PAGE gels and transferred to nitrocellulose membranes. After blocking with 5% BSA in Tris-buffered saline with Tween-20, membranes were incubated with primary (anti-BiP 1:1,000, anti-GRP94 1:5,000 and anti-HA 1:5,000 in blocking buffer) and secondary antibody solutions (goat anti-mouse or anti-rabbit HRP-coupled secondary antibodies 1:20,000) and finally developed using a BioRad ChemiDoc Imager.

## Statistics and reproducibility

SDS–PAGE gel bands were quantified in ImageLab (BioRad) and the data were analyzed and plotted in Prism (GraphPad). All quantifications were derived from three independent biological replicates with the mean and s.d. shown in the respective figures. Analytical size-exclusion experiments were performed at least twice and yielded consistent results for all runs (examples in Figs. 1b,c and 5c and Extended Data Figs. 1a and 2b). The combined crosslinking–SEC experiment shown in Extended Data Fig. 1b was performed once. Chemical crosslinking and the subsequent SDS–PAGE analysis presented in Extended Data Fig. 2a,c,d was performed at least twice and gave consistent results. The limited proteolysis of HT2 shown in Fig. 3b was repeated more than three times (with varying time points) and yielded consistent results. The representative CD spectroscopy shown in Fig. 3c was repeated twice for samples treated with HyT36 with consistent results and once for each of the control conditions. The sequential pulldown shown in Fig. 3f was repeated twice under the optimized conditions described here with consistent results.

## Reporting summary

Further information on research design is available in the Nature Portfolio Reporting Summary linked to this article.

## Data availability

The MS proteomics data were deposited to the ProteomeXchange Consortium through the PRIDE[88] partner repository with the dataset identifier PXD059917. Negative-stain EM maps were deposited to the EM Data Bank under accession numbers EMD-19600 and EMD-19601. Publicly accessible structures used for molecular modeling and data analysis in this work were obtained from the PDB under accession numbers 2KHO, 2O1V, 5E84, 5E85, 5ULS, 6DWS, 7KW7 and 8TF0. Source data are provided with this paper.

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

## Acknowledgements

We would like to thank M.-A. Derichs, D. Agranovski and L. Sewald for supporting biochemical experiments. This work was funded by the Sofja Kovaleveskaja Award through the Alexander von Humboldt Foundation endowed by the Federal Ministry of Education and Research. D.H., L.B., L.R., F.K., M.K., S.P. and E.S.-G were supported by the Deutsche Forschungsgemeinschaft (DFG, German Research Foundation; SFB1430, project 424228829). E.S.-G. acknowledges instrumentation funding under the Großgeräteinitiative (project 436586093) and the support of Germany's Excellence Strategy (EXC-2033, project 390677874. Screening and acquisition of EM data were performed at the StruBiTEM Facility of the University of Cologne funded by the DFG (INST 216/949-1 FUGG) with support from M. Gunkel and E. Behrmann.

## Author contributions

J.C.B., L.B. and L.R. conducted all the biochemical experiments and prepared the MS samples. F.K. and M.K. processed the MS samples and analyzed the data together with J.C.B., L.B. and D.H. J.D., L.B. and L.R. performed the cell biology experiments. J.C.B and L.C.Z. prepared the samples for EM. L.C.Z. and S.P. performed the EM data collection, processing and analysis. Y.A.-H. carried out the structural modeling. Y.A.-H. and E.S.-G. designed, analyzed and wrote the results and discussion of the computational modeling. D.H. outlined the project and prepared the paper with input from all coauthors.

## Funding

## Competing interests

The authors declare no competing interests.

## Additional information

**Extended data** is available for this paper at https://doi.org/10.1038/s41594-025-01619-0.

**Correspondence and requests for materials** should be addressed to Doris Hellerschmied.

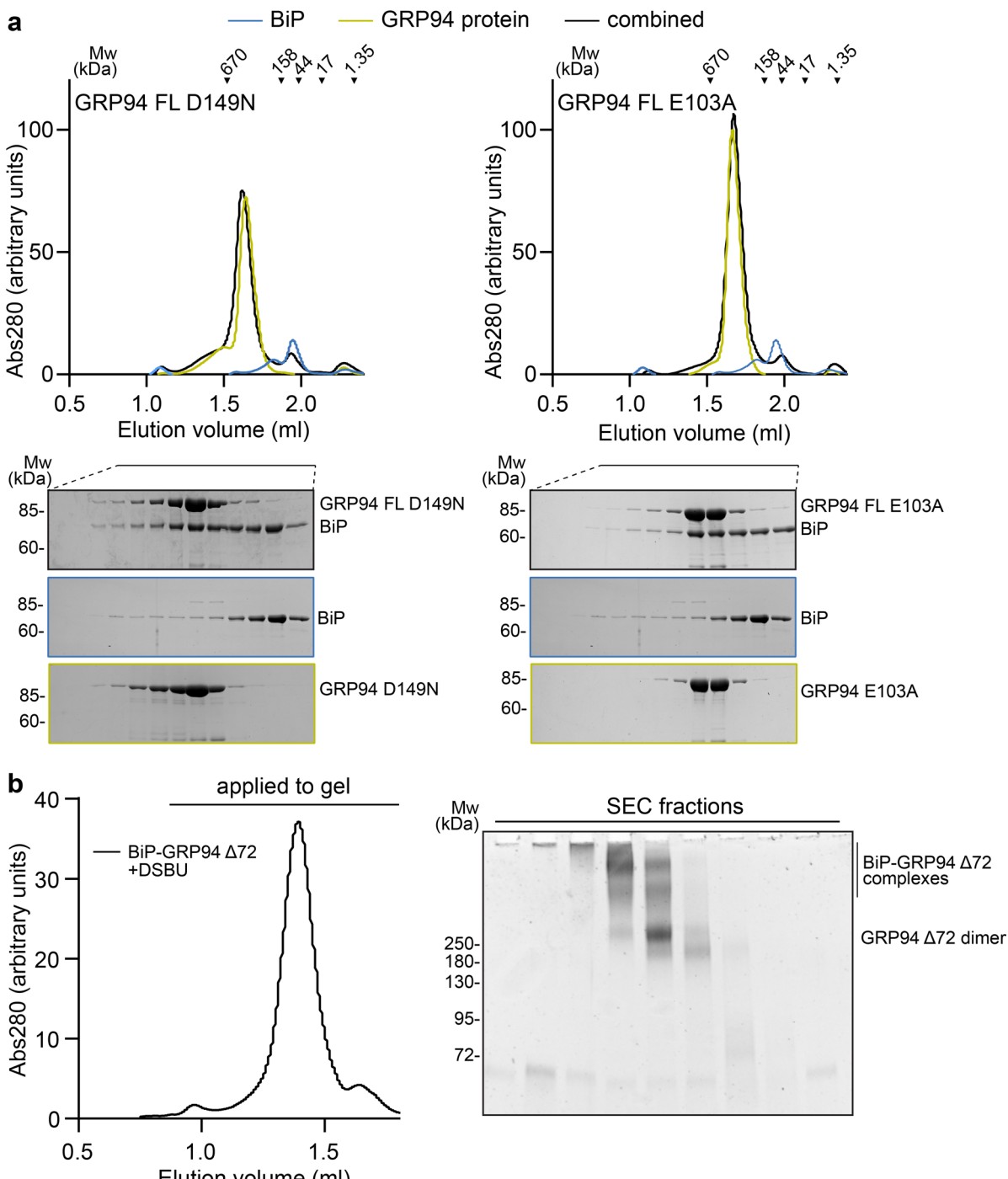

**Extended Data Fig. 1 | Additional biochemical data on the interaction of BiP and GRP94.** (**a**) Analytical SEC and SDS-PAGE analysis of complex formation between GRP94 full-length (FL) ATP-binding mutant D149N, ATP hydrolysis mutant E103A and BiP. (**b**) Analytical SEC and SDS-PAGE analysis of a BiP-GRP94 Δ72 sample crosslinked with DSBU. Protein bands on SDS-PAGE gels were visualized by Coomassie staining.

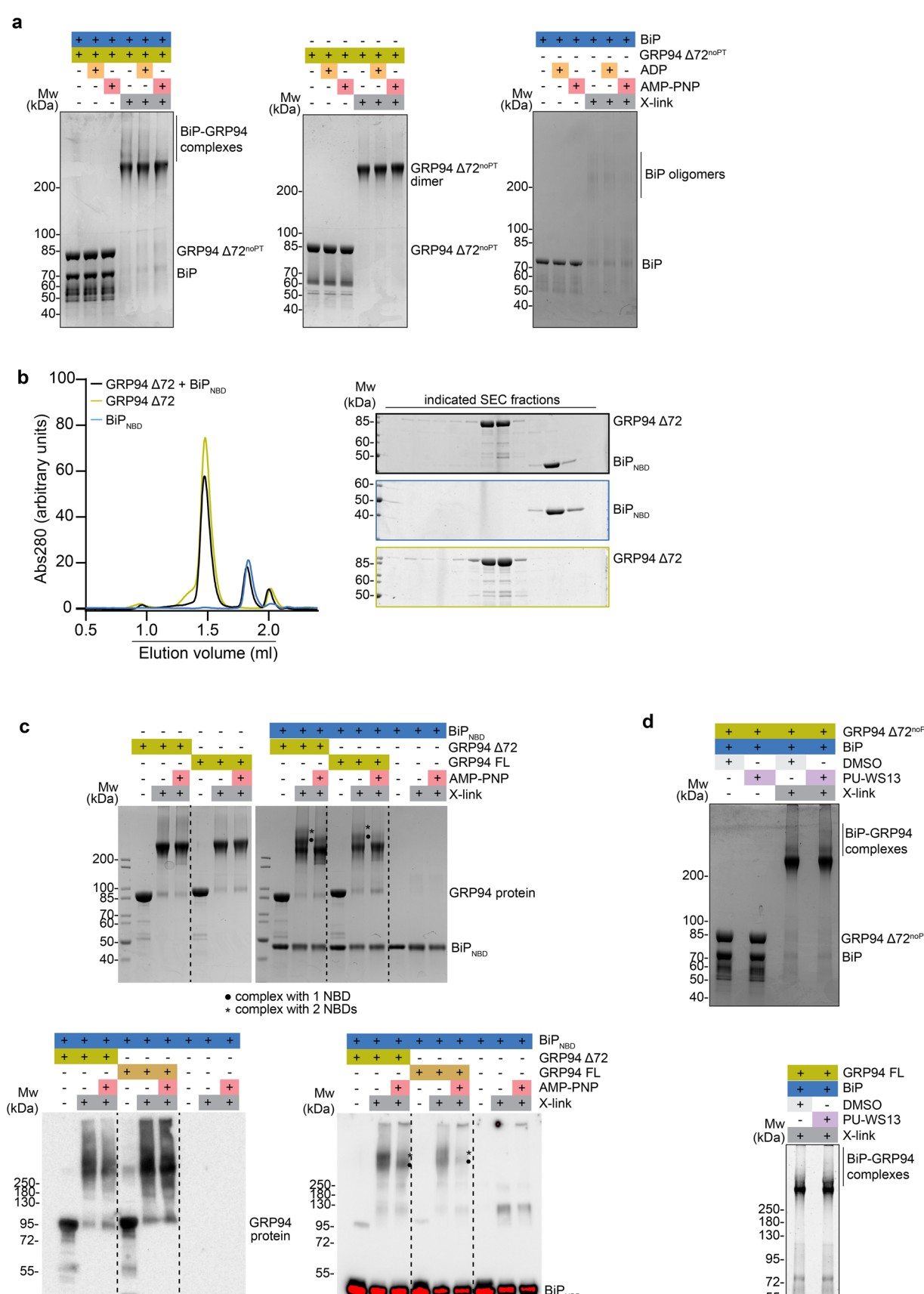

**Extended Data Fig. 2 | See next page for caption.**

**Extended Data Fig. 2 | Additional chemical crosslinking data on the interaction of BiP and GRP94.** (**a**) Chemical crosslinking of GRP94 Δ72$^{noPT}$ and BiP in the presence of ADP or AMP-PNP. Protein bands were visualized by Coomassie staining. (**b**) Analytical SEC and SDS-PAGE analysis of BiP$_{NBD}$-GRP94 Δ72 complex formation. Protein bands were visualized by Coomassie staining. (**c**) *Top panel:* SDS-PAGE of glutaraldehyde crosslinking reactions containing GRP94 Δ72 or FL protein and BiP$_{NBD}$ in the presence and absence of AMP-PNP. Protein bands were visualized by Coomassie staining. *Bottom panel:* Western blot analysis of glutaraldehyde crosslinking reactions to detect Strep-tagged GRP94 Δ72 or -FL protein and His$_6$-tagged BiP$_{NBD}$. (**d**) Chemical crosslinking of GRP94 Δ72$^{noPT}$ (top panel) or GRP94 FL (bottom panel) and BiP$_{NBD}$ in the presence of PU-WS13. Protein bands were visualized by Coomassie staining.

**a**

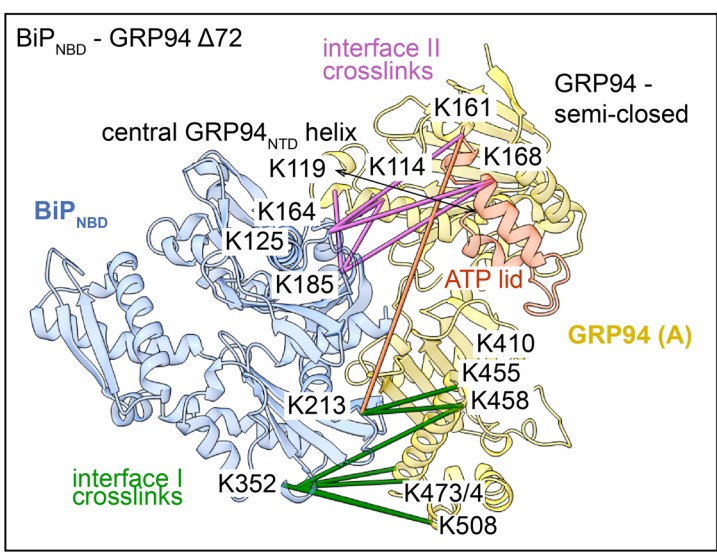

**c**

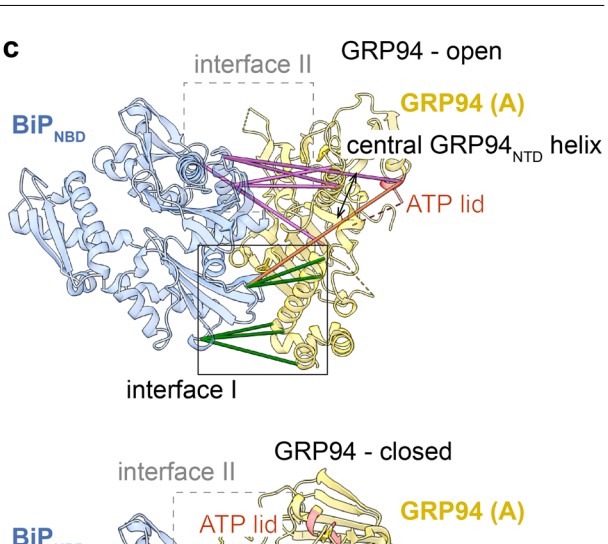

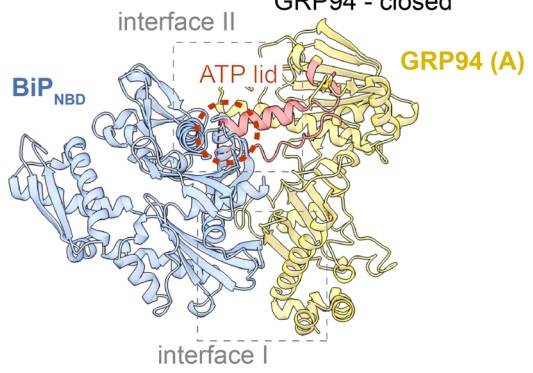

**b** crosslink categories:

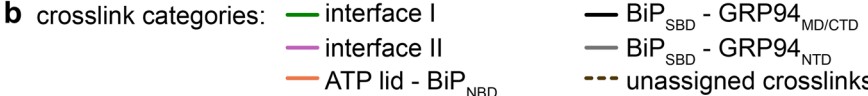

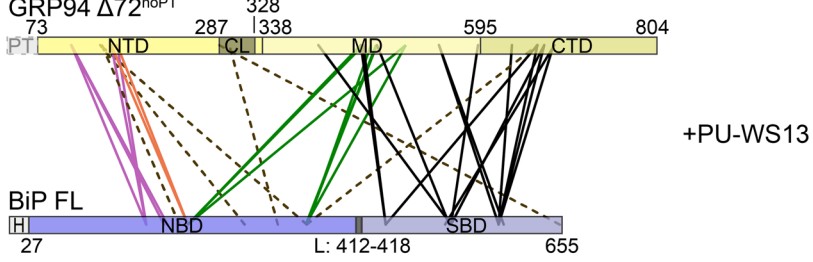

+PU-WS13

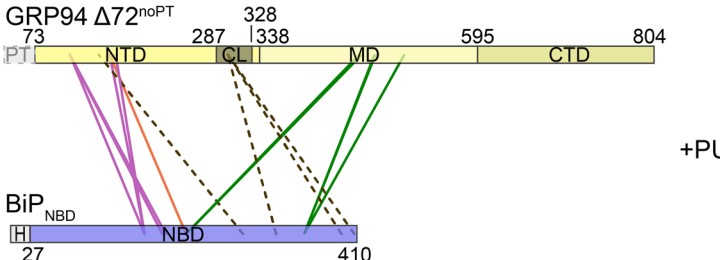

+PU-WS13

**Extended Data Fig. 3 | Additional analysis of BiP-GRP94 crosslinks.**
(**a**) Crosslinks outlining interface I, interface II, and the ATP lid – BiP$_{NBD}$ interactions from the BiP-GRP94 Δ72 XL-MS dataset mapped onto the homology model of BiP-GRP94 on the basis of the cytosolic loading complex (PDB ID: 7KW7). (**b**) Map of crosslinks identified between GRP94 Δ72$^{noPT}$ and BiP (top panel) or the BiP$_{NBD}$ (bottom panel) in the presence of the GRP94 ATPase inhibitor PU-WS13. NTD: N-terminal domain, MD: middle domain, CTD:

C-terminal domain, NBD: nucleotide-binding domain, SBD: substrate binding domain, PT: purification tag (which was removed by TEV cleavage), CL: charged linker, L: linker, H: His$_6$-tag. (**c**) Crosslinks outlining interface I, interface II, and the ATP lid – BiP$_{NBD}$ interactions from the BiP-GRP94 FL XL-MS dataset (comparable to Fig. 2c) mapped onto GRP94 protomer A in the open (2O1V-chain A) and closed (5ULS-chain A) conformation aligned to the semi-closed conformation model shown in Fig. 2b.

**a** GRP94 - open

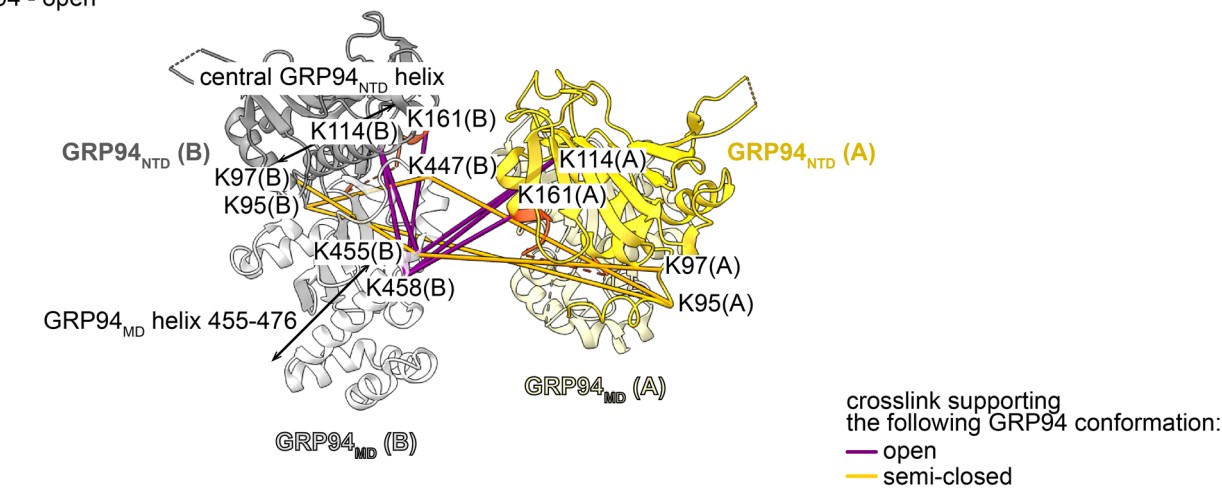

GRP94 - semiclosed

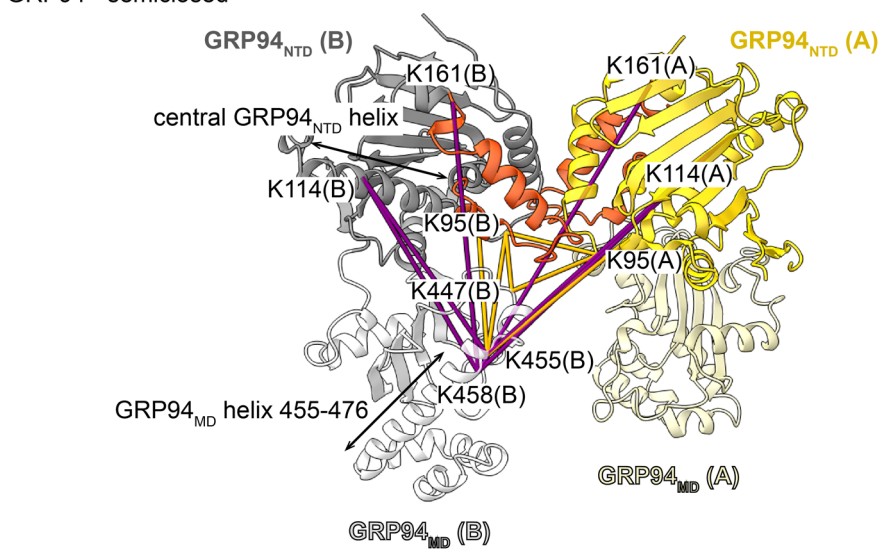

**b**

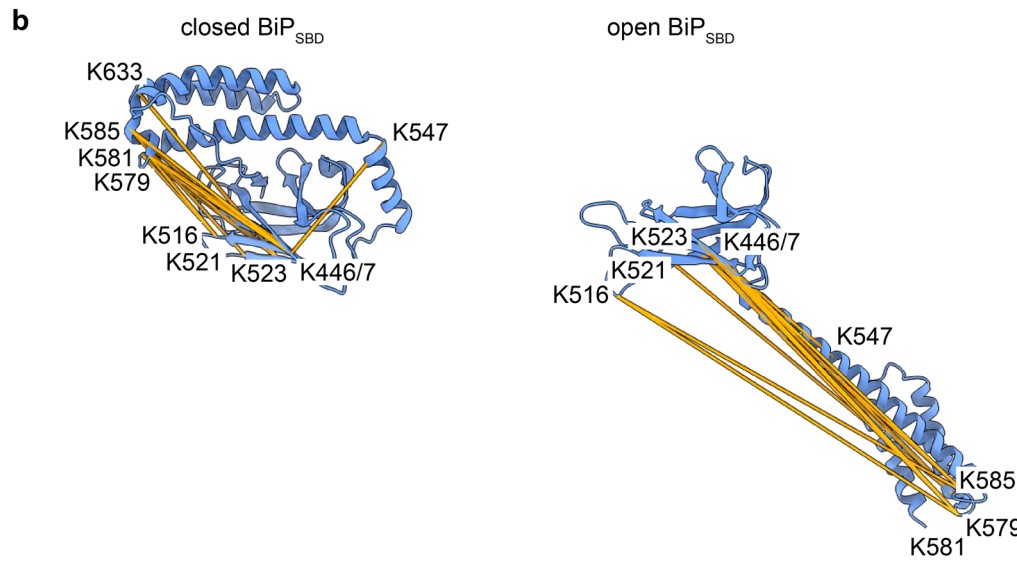

**Extended Data Fig. 4 | See next page for caption.**

**Extended Data Fig. 4 | Crosslinking mass spectrometry outlines the conformations of BiP and GRP94.** (**a**) Intra-GRP94 crosslinks mapped onto the open (PDB ID: 2O1V) or semi-closed (homology model based on PDB ID: 7KW7) GRP94 dimer. For simplicity, only MD and NTD are shown. Crosslinks from the MD region in protomer B - K455 (B), K458 (B), K447 (B) to the NTD region within the same protomer and to protomer A are shown - K114 (A/B), K161 (A/B), K95 (A/B), K97 (A/B). Crosslinks shown in purple support the open conformation, crosslinks shown in orange support the semi-closed conformation. (**b**) Intra-

$BiP_{SBD}$ crosslinks between the SBDα and SBDβ (identified in the BiP-GRP94 FL XL-MS dataset, Supplementary Table 1) mapped onto the closed (left panel, PDB ID: 5E85) and open (right panel, PDB ID: 5E84) $BiP_{SBD}$ structure. The $BiP_{SBD}$ domains are aligned based on the SBDβ. Crosslinks between the beta-sheet-rich subdomain, SBDβ (residues K446, K447, K521, K523) and the alpha-helical lid, SBDα (residues K573, K579, K581, K585) outline their close proximity and therefore indicate that the $BiP_{SBD}$ is in the closed conformation.

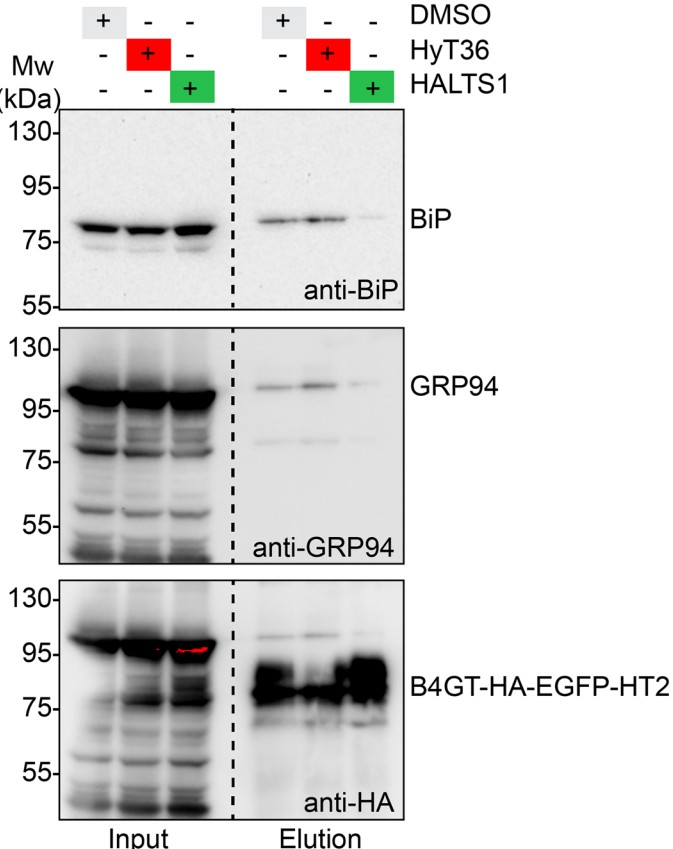

**Extended Data Fig. 5 | EGFP-trap pull-down of B4GT-HA-EGFP-HT2 from HEK293 cells.** The Golgi and ER-localized B4GT-HA-EGFP-HT2 fusion protein was pulled down from HEK293 cells. Western blot analysis of input and elution fractions probing for the bait protein, BiP, and GRP94 is shown. HyT36-treatment increased the amounts of BiP and GRP94 pulled down with B4GT-HA-EGFP-HT2, while treatment with the small molecule HALTS1, a HT2-stabilizer, decreased chaperone binding demonstrating that BiP and GRP94 interact with HT2 fusion proteins in cells, depending on the HT2 folding state.

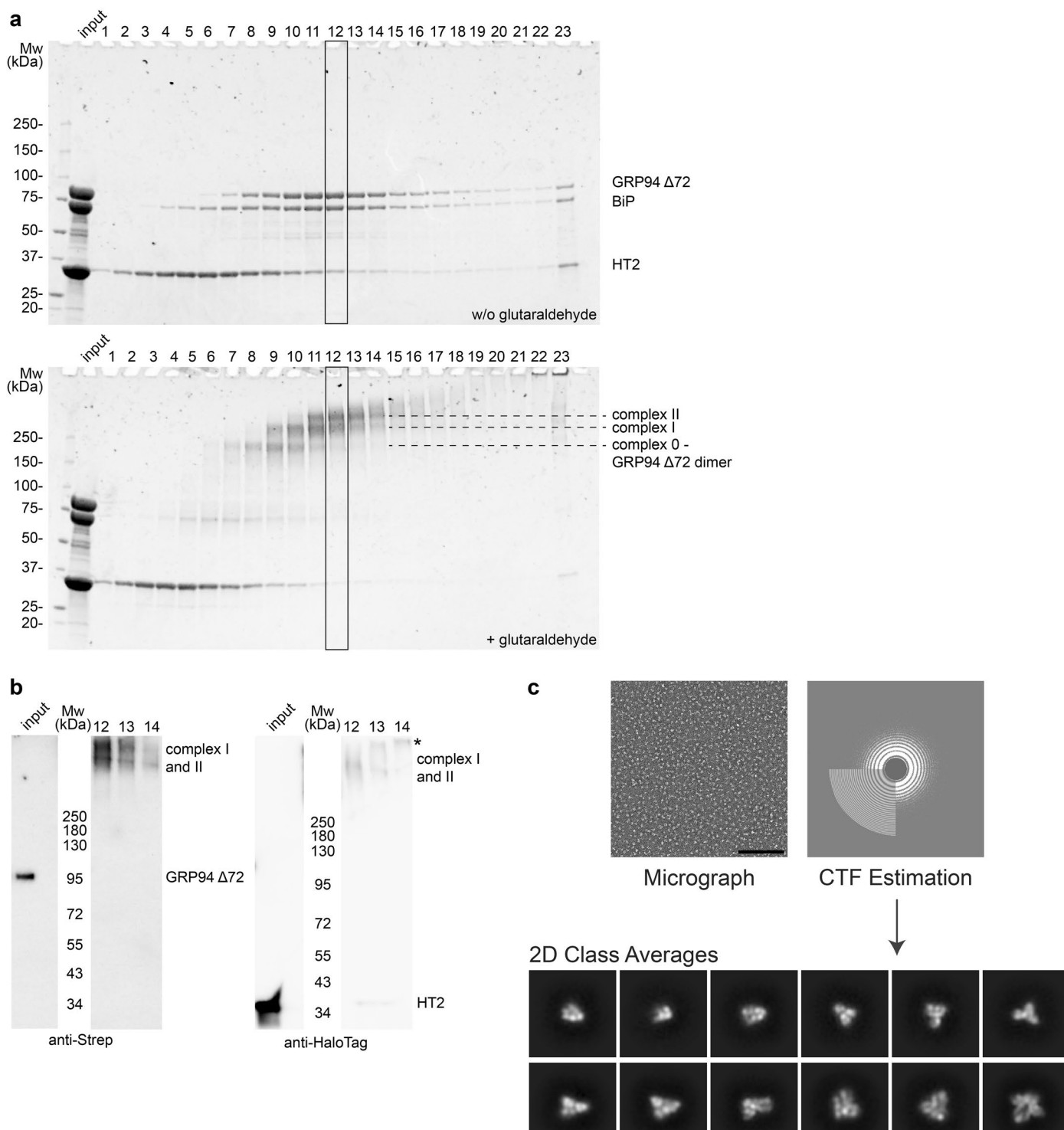

**Extended Data Fig. 6 | Characterization of negative-stain EM sample. (a)** SDS-PAGE gel analysis of GraFix fractions. Gradient density increases with fraction number. Fraction 12 was analyzed by negative-stain EM. Protein bands were detected by Coomassie staining. **(b)** Western blot analysis of input and gradient fractions 12, 13, and 14 containing glutaraldehyde. The use of different gels accounts for the different banding pattern compared to **(a)**. The asterisk denotes potentially aggregated HT2 protein. **(c)** Representative negative-stain EM micrograph and 2D class averages obtained from fraction 12. Scale bar represents 200 nm.

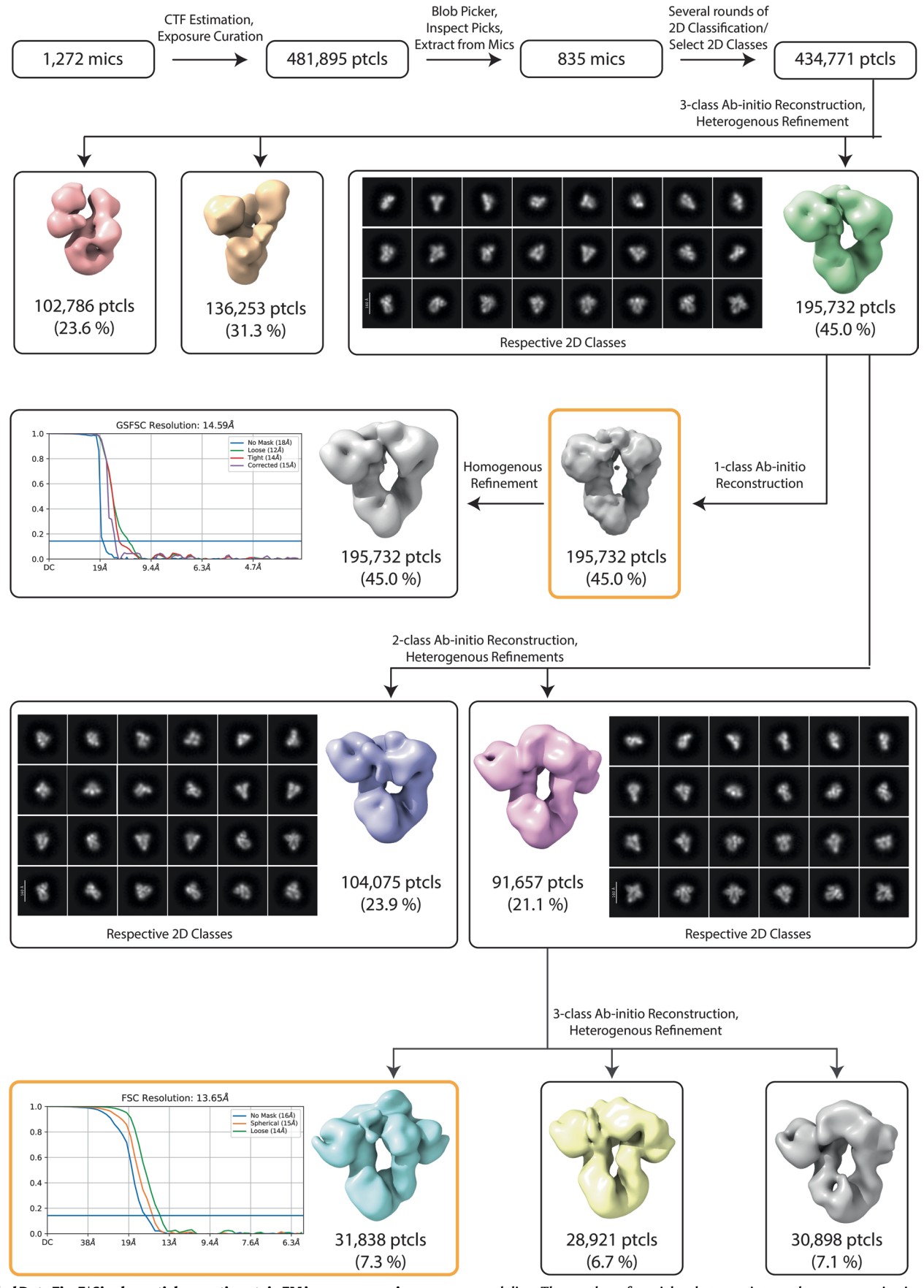

**Extended Data Fig. 7 | Single particle negative-stain EM image processing pipeline.** Flow chart of classifications and refinements, resulting in reconstructions of the pre-loading and loading complex conformations. Yellow boxes indicate 3D reconstructions that were further used for molecular modeling. The number of particles that went into each reconstruction is indicated, together with their respective percentage share of the initial 434,771 particles. 2D classifications of those particle stacks resulted in the shown respective 2D class averages (scale bar: 160 Å).

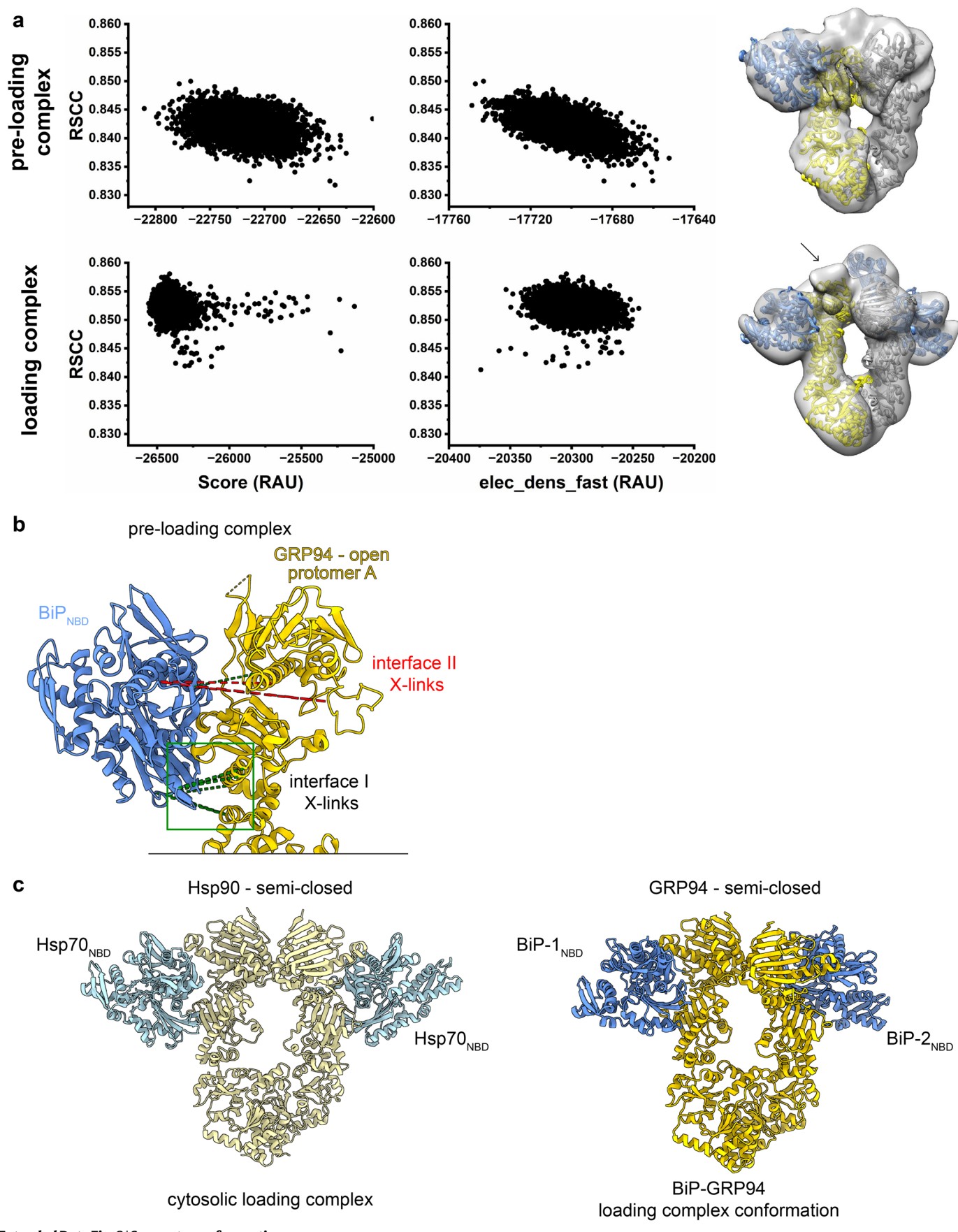

**Extended Data Fig. 8 | See next page for caption.**

**Extended Data Fig. 8 | Evaluation of pre-loading and loading complex models.** (**a**) Quality metrics of molecular modeling of the pre-loading and loading complexes. RSCC: Real-space correlation coefficient, RAU: Rosetta arbitrary units. The arrow indicates additional density not filled by the loading complex models. (**b**) Crosslinks identified between BiP$_{NBD}$ and GRP94 Δ72 mapped onto the pre-loading complex model. GRP94 protomer B is omitted for clarity. Red lines indicate crosslinks > 30 Å, green lines < 30 Å. (**c**) Side-by-side comparison of the cytosolic (left) and BiP-GRP94 (right) loading complex conformation. For clarity, exclusively the NBDs of Hsp70/BiP and the Hsp90/GRP94 dimer structures are shown.

**a**

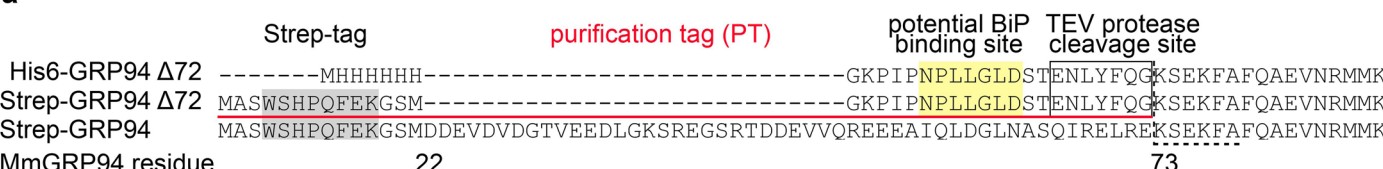

**b**

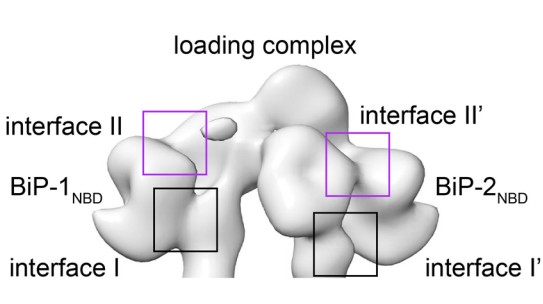

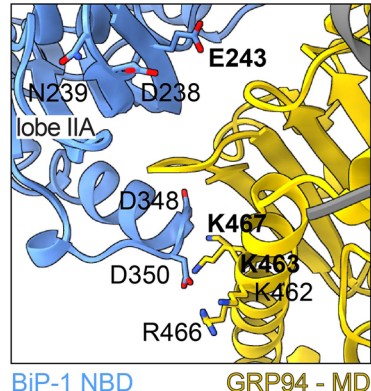

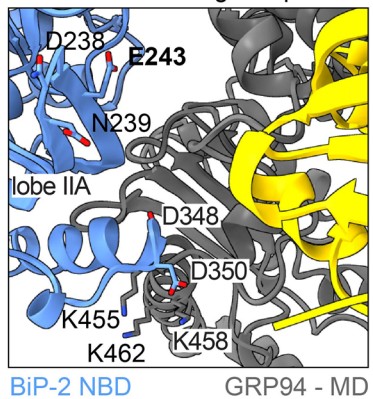

**Extended Data Fig. 9 | BiP-GRP94 interaction interfaces in the loading complex conformation.** (**a**) Sequence alignment of the indicated expression constructs. A sequence stretch that is part of the His₆- and Strep-tagged GRP94 Δ72 constructs highlighted in yellow may serve as a BiP substrate mimetic. (**b**) Zoom-in into interfaces I, I' and II' of the BiP-GRP94 loading complex. Compare to Fig. 5b and e.

**Extended Data Table 1 | Negative-stain EM data**

| | GRP94-BiP-HT2 Pre-Loading Complex (EMD-19600) | GRP94-(BiP)$_2$-HT2 Loading Complex (EMD-19601) |
|---|---|---|
| **Data collection and processing** | | |
| Microscope | Talos L120C | Talos L120C |
| Detector | Ceta 16M | Ceta 16M |
| Voltage (keV) | 120 | 120 |
| Electron exposure (e–/Å$^2$) | ~25 | ~25 |
| Magnification | 73,000 | 73,000 |
| Defocus range (µm) | 0.7 – 2.2 | 0.7 – 2.2 |
| Pixel size (Å) | 1.89 | 1.89 |
| Symmetry imposed | C1 | C1 |
| Number of particles | 195,732 | 31,838 |
| Map resolution (Å) | 15 | 14 |
| FSC threshold | 0.143 | 0.143 |

| | |
|---|---|

# Reporting Summary

## Statistics

For all statistical analyses, confirm that the following items are present in the figure legend, table legend, main text, or Methods section.

| n/a | Confirmed | |
|---|---|---|
| ☐ | ☒ | The exact sample size (*n*) for each experimental group/condition, given as a discrete number and unit of measurement |
| ☐ | ☒ | A statement on whether measurements were taken from distinct samples or whether the same sample was measured repeatedly |
| ☒ | ☐ | The statistical test(s) used AND whether they are one- or two-sided<br>*Only common tests should be described solely by name; describe more complex techniques in the Methods section.* |
| ☒ | ☐ | A description of all covariates tested |
| ☒ | ☐ | A description of any assumptions or corrections, such as tests of normality and adjustment for multiple comparisons |
| ☐ | ☒ | A full description of the statistical parameters including central tendency (e.g. means) or other basic estimates (e.g. regression coefficient) AND variation (e.g. standard deviation) or associated estimates of uncertainty (e.g. confidence intervals) |
| ☒ | ☐ | For null hypothesis testing, the test statistic (e.g. *F*, *t*, *r*) with confidence intervals, effect sizes, degrees of freedom and *P* value noted<br>*Give P values as exact values whenever suitable.* |
| ☒ | ☐ | For Bayesian analysis, information on the choice of priors and Markov chain Monte Carlo settings |
| ☒ | ☐ | For hierarchical and complex designs, identification of the appropriate level for tests and full reporting of outcomes |
| ☒ | ☐ | Estimates of effect sizes (e.g. Cohen's *d*, Pearson's *r*), indicating how they were calculated |

*Our web collection on statistics for biologists contains articles on many of the points above.*

## Software and code

Policy information about availability of computer code

| Data collection | MS data were collected on an Orbitrap LUMOS instrument (Thermo) coupled to an EASY-nLC 1200 ultra-performance liquid chromatography (UPLC) system (Thermo). Negative-stain EM data were collected on a Talos L120C microscope (Thermo Fisher) with a CMOS camera (Ceta 16M). |
|---|---|
| Data analysis | Negative-stain EM data was processed in CryoSPARC v4, using CTFFIND4 for CTF estimation and Blob picker for particle picking. XL-MS data were processed with ProteoWizard (version: 3.0.23018-60066e9), Xisearch (version: 1.7.6.7), and xiFDR (version 2.1.5.5). For molecular modelling and analysis of structural data SWISS-MODEL web server (accessed 13.10.2023), UCSF Chimera (version 1.17.3), UCSF Chimera X (version 1.5), Rosetta (release 314, 2022.11+release.512e589) , and PyMOL 2.5.7 were used. For image quantification ImageLab 6.1 (BioRad) and Prism 10.1.2 (Graphpad) were used. |

For manuscripts utilizing custom algorithms or software that are central to the research but not yet described in published literature, software must be made available to editors and reviewers. We strongly encourage code deposition in a community repository (e.g. GitHub). See the Nature Portfolio guidelines for submitting code & software for further information.

## Data

Policy information about **availability of data**

All manuscripts must include a **data availability statement**. This statement should provide the following information, where applicable:

- Accession codes, unique identifiers, or web links for publicly available datasets
- A description of any restrictions on data availability
- For clinical datasets or third party data, please ensure that the statement adheres to our **policy**

> The mass spectrometry proteomics data have been deposited to the ProteomeXchange Consortium via the PRIDE partner repository (https://www.ebi.ac.uk/pride/archive/) with the dataset identifier PXD059917.
> Negative-stain EM maps have been deposited in the Electron Microscopy Data Bank (EMDB) under the accession numbers EMD-19600 and EMD-19601. The following publicly accessible PDB models were used for molecular modeling and data analysis in this work: 2KHO, 2O1V, 5E84, 5E85, 5ULS, 6DWS, 7KW7, 8TF0.

## Research involving human participants, their data, or biological material

Policy information about studies with **human participants or human data**. See also policy information about **sex, gender (identity/presentation), and sexual orientation** and **race, ethnicity and racism**.

| | |
|---|---|
| Reporting on sex and gender | n/a |
| Reporting on race, ethnicity, or other socially relevant groupings | n/a |
| Population characteristics | n/a |
| Recruitment | n/a |
| Ethics oversight | n/a |

Note that full information on the approval of the study protocol must also be provided in the manuscript.

# Field-specific reporting

Please select the one below that is the best fit for your research. If you are not sure, read the appropriate sections before making your selection.

☒ Life sciences ☐ Behavioural & social sciences ☐ Ecological, evolutionary & environmental sciences

For a reference copy of the document with all sections, see nature.com/documents/nr-reporting-summary-flat.pdf

# Life sciences study design

All studies must disclose on these points even when the disclosure is negative.

| | |
|---|---|
| Sample size | Sample sizes for biochemical experiments were chosen based on common practice in the field to account for experimental and measurement variation and to ensure reproducible results: *) Analytical size exclusion chromatography runs, limited proteolysis, and crosslinking experiments were repeated at least twice and gave reproducible and consitent results. *) For quantifications of pull-down and crosslinking experiments three independent biological replicates were performed (as reported in the figure legends). Particle numbers for single-particle negative-stain EM are described in detail in Extended Data Figure, including a flow chart indicating the number and percentages of particles that were used for each corresponding 3D reconstruction. |
| Data exclusions | No data was excluded. |
| Replication | The number of replications is stated in the figure legends and/or in the Statistics and Reproducibility statement. |
| Randomization | Randomization was not applicable, since experiments did not involve humans or animals. Moreover, the scientist performing the biochemical experiments needed to be aware of which proteins have to be added to the tubes. |
| Blinding | No blinding was performed, since no subjective analysis was performed. |

# Reporting for specific materials, systems and methods

We require information from authors about some types of materials, experimental systems and methods used in many studies. Here, indicate whether each material, system or method listed is relevant to your study. If you are not sure if a list item applies to your research, read the appropriate section before selecting a response.

## Materials & experimental systems

| n/a | Involved in the study |
|-----|----------------------|
| ☐ | ☒ Antibodies |
| ☐ | ☒ Eukaryotic cell lines |
| ☒ | ☐ Palaeontology and archaeology |
| ☒ | ☐ Animals and other organisms |
| ☒ | ☐ Clinical data |
| ☒ | ☐ Dual use research of concern |
| ☒ | ☐ Plants |

## Methods

| n/a | Involved in the study |
|-----|----------------------|
| ☒ | ☐ ChIP-seq |
| ☒ | ☐ Flow cytometry |
| ☒ | ☐ MRI-based neuroimaging |

## Antibodies

| | |
|---|---|
| Antibodies used | anti-His5 (Santa Cruz Biotechnologies, SC-8036), anti-Strep (Qiagen, 34850), anti-BiP C50B12 (Cell Signaling, 3177), anti-GRP94 (Proteintech, 14700-1-AP), anti-HA-tag C29F4 (Cell Signaling 3724), anti-HaloTag (Promega, G9211), Goat anti-mouse HRP-coupled secondary antibody (Invitrogen, 31444), and Goat anti-rabbit HRP-coupled secondary antibody (Invitrogen, 31460) |
| Validation | The anti-His5, anti-Strep, anti-HaloTag, and anti-HA antibodies were used to detect tags on purified proteins or proteins overexpressed in HEK293 cells. No band was detected when the corresponding tagged protein was omitted from the sample, validating the use of these antibodies in our experimental setup. Specifically, Extended Data Figure 2c and 6b show that the anti-Strep antibody recognizes Strep-tagged GRP94. No band was detected for samples not containing any Strep-GRP94 protein (see Extended Data Figure 2c – lanes containing BiPNBD only). Extended Data Figure 6b also shows that the anti-HaloTag antibody specifically detects a single band at the expected molecular weight for HaloTag2 and does not react with BiP or GRP94, which are also present in the input sample. The anti-His5 antibody detects the BiPNBD at the expected molecular weight as demonstrated in Extended Data Figure 2b. |
| | The anti-HA tag antibody was used to detect B4GT-HA-EGFP-HT2, which is expressed upon Doxycycline induction in HEK293 cells. As demonstrated in Hellerschmied et al MBoC 2019, Figure 1D, the antibody specifically recognizes the HA-tagged protein upon addition of Doxycycline. |
| | Anti-BiP and anti-GRP94 antibodies were approved for use in Western blot experiments by their respective manufacturers. ProteinTech provides Western blot proof of function for the anti-GRP94 antibody in HEK293 cells (used in this work) and a GRP94 knock-down validation (in prostate cancer cells) is for example provided in this publication 10.1515/biol-2019-0043 https://www.ptglab.com/products/HSP90B1-Antibody-14700-1-AP.htm Cell signaling provides proof of function for Western blot and showcases over 900 publications using the anti-BiP (3177) antibody, as listed on the following website https://www.cellsignal.com/products/primary-antibodies/bip-c50b12-rabbit-mab/3177 |

## Eukaryotic cell lines

Policy information about cell lines and Sex and Gender in Research

| | |
|---|---|
| Cell line source(s) | Stable HEK293 Flp-in Trex cells (Thermo Fisher Scientific) expressing B4GT-HA-EGFP-HT2 were previously generated and described in Serebrenik et al., 2018. |
| Authentication | The HEK293 Flp-in Trex cell line used was not authenticated. |
| Mycoplasma contamination | The cell line tested negative for mycoplasma contamination using the VenorGeM OneStep mycoplasma detection kit. |
| Commonly misidentified lines (See ICLAC register) | No commonly misidentified cell line was used in this work. |

## Plants

| | |
|---|---|
| Seed stocks | *Report on the source of all seed stocks or other plant material used. If applicable, state the seed stock centre and catalogue number. If plant specimens were collected from the field, describe the collection location, date and sampling procedures.* |
| Novel plant genotypes | *Describe the methods by which all novel plant genotypes were produced. This includes those generated by transgenic approaches, gene editing, chemical/radiation-based mutagenesis and hybridization. For transgenic lines, describe the transformation method, the number of independent lines analyzed and the generation upon which experiments were performed. For gene-edited lines, describe the editor used, the endogenous sequence targeted for editing, the targeting guide RNA sequence (if applicable) and how the editor was applied.* |
| Authentication | *Describe any authentication procedures for each seed stock used or novel genotype generated. Describe any experiments used to assess the effect of a mutation and, where applicable, how potential secondary effects (e.g. second site T-DNA insertions, mosiacism, off-target gene editing) were examined.* |

