## [Peer Review File · Nature Structural & Molecular Biology]

Conformational plasticity of a BiP-GRP94 chaperone complex

Corresponding Author: Professor Doris Hellerschmied

A version of this paper was originally rejected for publication by Nature Structural & Molecular Biology, however that decision was reconsidered after appeal by the authors.

Version 0:

Decision Letter:

21st Mar 2024

Dear Dr. Hellerschmied,

Thank you for submitting your manuscript "Conformational plasticity of a BiP-GRP94 chaperone complex". The comments from the 3 reviewers who have evaluated your manuscript are below. Unfortunately, after carefully considering their comments, we cannot offer to publish your manuscript in Nature Structural & Molecular Biology.

You will see that while the referees find the work of potentially interesting, they raise concerns about the strength of the novel conclusions that can be drawn at this stage.

However, if further experimentation, analysis, and revisions allow you to address the referees concerns in full, we would be prepared to consider an appeal of our decision, on the condition that no related work is published in the interim or has been accepted in our journal. Please contact me to discuss an appeal and potential revision. Please note that, until we have the opportunity to read the revised manuscript in its entirety, we cannot promise that it will be sent back for peer review.

I am sorry we could not be more positive on this occasion. I hope that you find the referees' comments useful in deciding how best to proceed.

Sincerely,

Katarzyna Ciazynska, PhD
(she/her)
Associate Editor
Nature Structural & Molecular Biology
<https://orcid.org/0000-0002-9899-2428>

Referee expertise:

Referee #1: chemical biology

Referee #2: XL-MS

Referee #3: protein folding

Reviewers' Comments:

Reviewer #1:

Remarks to the Author:

Brenner et al provide a biochemical and structural analysis of the interaction between the chaperones BiP and GRP94, which are critical for the correct folding of proteins in the secretory pathway. These two proteins cooperate but the structural understanding of these chaperoning machines and their interplay has been limited by their high conformational flexibility.

The key findings of the manuscript are (i) that BiP and GRP94 may form a complex very similar to the loading complex previously described for the cytosolic paralogs Hsp70 and Hsp90, (ii) that to get there, BiP and GRP94 don't seem to need the assistance of cochaperones like Hop, which is essential for the assembly of Hsp70-Hsp90 complexes, and (iii) the identification and structural description of a pre-loading complex that may precede the observed loading complexes. These observations are important and will be influential to the chaperone field. However, the manuscript could improve substantially by solidifying the relevance of the observations as well as by some minor amendments.

Major issues:

1. Did the author also test a GRP94 $\Delta 72$, where the purification tag was removed (other than Fig. 4f)? This is important as the tag might have been crucial for stabilized the observed complex. But its stabilization might also be non-physiological.
2. Why did the authors use the BiPNBD construct in the chemical (glutaraldehyde) crosslinking when it didn't bind on its own. The authors should repeat this with the BiP-GRP94 $\Delta 72$ constructs used later for structural studies
3. The authors should test the effect of an GRP94 inhibitor on complex formation (e.g. the chemical (glutaraldehyde) crosslinking) which would be expected to block the closed conformation
4. The XL-MS experiments need control experiments and a much better representation/explanation:
 - a) Fig. 3a: Use different colors (for the different types of inter-BiP-GRP94 crosslinks) to indicate, which crosslinks correspond to interface I, interface II, and to the purification tag. Why do the authors observe so much more crosslinks for the GRP94-NTD/BiP-SBD and GRP94-PT/BiP-SBD compared to the interfaces I and II, which are at the heart of this manuscript.
 - b) 'The observed crosslinks between BiP and GRP94 $\Delta 72$ suggest that the two chaperones adopt a similar conformation as their cytosolic paralogs, Hsp70 and Hsp90, in the previously described loading complex.' Please amend ED Tab1. to include a comment whether the particular crosslinks are compatible (or not) with the homology modeled based on PDB-ID 7kw7. Please also include, which distance cutoff will be used for this assessment.
 - c) Please repeat this analysis with wildtype GRP94 and BiP – I consider this much more relevant than the experiment with/without HT2. It helps to support the proposed structures in the absence of the (potentially artificial) stabilization by the tag.

Minor issues:

5. I hardly see a difference in BiP co-elution percentage between GRP94wt vs D149N vs E103A (Fig. 1b & ED1b). If there was a 'minor increase in BiP co-eluting with GRP94 D149N, why did the author not use this mutant later on (I assume the following experiments were performed with GRP94 containing a wildtype NTD).
6. 'To further assess the effect of the GRP94 nucleotide-bound state on complex formation we used a BiPNBD construct, which irrespective of its nucleotide state, retains GRP94 binding' Please explain why the authors used this construct and GRP94 binding is irrespective of its nucleotide state here
7. '...with an even more pronounced band corresponding to a 133 GRP94 $\Delta 72$ dimer with two BiPNBD molecules bound' I find the evidence for two BiPNBD molecules not that strong; the bands are rather weak compared to the crosslinked GRP94 dimer alone
8. The authors work on the HT2-tag system is beautiful. It is unfortunate that the HT2 substrate could not be detected in the XL-MS or the cryo-EM structures. The lack of well defined and tractable authentic substrates for BiP/GRP94 is a major limitation for this branch of the chaperone field.
9. 'we performed Gradient Fixation (GraFix), which resulted in the separation of three major complex species' I don't see a real separation of the complex species (by the GraFix procedure)
10. 'Upon removal of the GRP94 purification tag, we observed a reduction in GRP94 protein bound to BiP by ~65%, suggesting that the tag contributes to the stability of the chaperone complex.' Could this represent primarily a complex with only one BiP bound compared to a general destabilization of the complex?
11. 'We presume that interactions between substrate protein and GRP94 drive the formation of the loading complex, in addition to interactions between the GRP94NTDs and BiPNBDs, further facilitating substrate transfer.' Please discuss how these processes might be coupled
12. 'our structural and biochemical data provide insight into the BiP-GRP94 pre-loading and loading complex conformations, two successive states along the GRP94 chaperone cycle'

This is indeed tempting to postulate – do the author have any data for supporting a transformation of pre-loading to loading complex? Perhaps this should be phrase a bit more cautiously.

13. Please address in the discussion (and in Fig. 6) – where possible – the role of ADP/ATP states and ATP hydrolysis/nucleotide release in the chaperoning cycle

Reviewer #2:

Remarks to the Author:

The manuscript by Brenner et al. describes a combination of crosslinking mass spectrometry (XL-MS) and negative-stain electron microscopy (EM) for the analysis of GRP94-BiP chaperone complexes. By using hydrophobic tagging to include an unfolded substrate in the reconstituted system, the authors were able to isolate the complex in the pre-loading and loading complex states and both biochemically and structurally characterize these forms. They identify critical interaction sites between GRP94 and BiP and compare these interactions with those for HSP90 and HSP70. Overall, the XL-MS and EM results are in good agreement and provide unique insights into the conformations.

Major Comments:

1. The authors assume that intramolecular crosslinks in GRP94 are generally interprotomer rather than intraprotomer, as assessed from the indicated crosslinks in Figure 3. How did the authors determine which of the crosslinks shown in Figure 3 are intraprotomer and which were interprotomer? Although these come from high molecular weight bands, can the authors necessarily exclude that the intraprotomer crosslinks come from two different molecules rather than one? For example, were monomer crosslink bands also analyzed to see what intramolecular crosslinks form there to support that these are interprotomer? If not, then the results need to be amended to clarify that the assignment of intraprotomer or interprotomer is hypothetical and that an alternate assignment is possible.
2. Were any other crosslinkers used to attempt to get some crosslinks with HT2? Do the authors have any thoughts on whether the lack of HT2 crosslinks is due to technical challenges of the experiment itself (no good crosslinking results in HT2 proximal to where it might bind, for example) or instability in the complex that prevents such crosslinks from readily forming and being detected?
3. Tables of the intramolecular crosslinks in the Extended Data should be included.
4. Are the crosslinks of BiP with the GRP94d72 purification tag consistent with the stabilization effect observed?
5. Were the XL-MS experiments repeated on separately-prepared samples, or do these represent results from a single set of samples?

Minor Comments:

6. In Extended Data Table 1, it seems that the scores from Xisearch are color coded, but that is not described.
7. Page 6, lines 214 and 217 talking about the crosslinks between the GRP94MD GRP94NTD refer to Fig 3b, should be 3c?

Reviewer #3:

Remarks to the Author:

This manuscript describes an in depth characterization of the interaction between the ER chaperones BiP and GRP94. The authors use size exclusion chromatography, chemical crosslinking combined with mass spectrometry, modeling studies, and negative stain electron microscopy to show that BiP has multiple interaction surfaces on GRP94. The authors also investigate the interaction between a model unfolded substrate, HT2, and the BiP/GRP94 complex and show that they can isolate a complex containing all three of the proteins.

Overall this is a technical tour de force. The formation of the ternary complex with HT2 is a terrific achievement. On the other hand, I have several reservations about the paper that should be considered before proceeding further:

First, while the in vitro complex with the model unfolded substrate HT2 is a novel result, it is not clear whether this reflects a biologically relevant state or activity for GRP94. The role of GRP94 in protein homeostasis and misfolding in the ER is, unfortunately, still very unclear. The authors cite references 46 and 47 (line 142) as evidence that GRP94 interacts with HT2 in HEK cells, but I was not able to find a convincing description of this specific interaction in either of these reports, which appear to focus mainly on the interaction with BiP. (GRP94 is not mentioned in either of these references, and the implication is that the KDEL proteins also identified include GRP94.) If GRP94 is not involved in misfolded protein correction or clearance, then the employment of the HT2 substrate is a clever approach to the substrate problem but ultimately artifactual. As such, the corollary is that these studies may not do much to illuminate the role of GRP94 in client maturation.

Second, despite demonstrating that HT2 is captured by BiP/GRP94 (Figure 2e, panel 2 is terrific), the authors are unable to further characterize this interaction. HT2 does not show up in any of their crosslinking studies, and it is not visible in the electron microscopy reconstructions. This is a quite surprising given the fact that the complex appears to be somewhat stable.

Third, while technically laudable, and clearly the product of a great deal of effort, the overall result of this study does not, unfortunately, add much to our understanding beyond what could have been achieved by homology modeling with the referenced Hsp90/Hsp70/GR complexes. It does not appear that any new interaction surfaces are identified, the location of the HT2 model substrate is not revealed, and there is also an artifactual BiP/GRP94-His tag interaction that may affect the

positioning of the BiP SBD in their models.

Detailed comments are listed below:

Line 107. In Figure 1b the BiP SEC line is hard to see. Perhaps a thicker line or different color could be used.

Line 117, 118. It would be helpful if the term “minor” could be quantified by giving a percent. These proportions are hard to determine in the Figure. Same as line 117 comment for the term “reduced.”

Line 131. I'm afraid that I wasn't able to see the “strongly reduced” complex the authors are referring to. Perhaps quantitation of the relevant supershifted complex bands would be helpful here.

Figure 1. The authors should indicate the method of detection used – staining or blot?

Line 135. “Protocol” might be replaced with something like “characterized the interaction.”

Line 142. References 46 and 47 do not seem to show GRP94 involvement with HT2 in cells.

Line 175. In Figure 2e it would be helpful to identify the band at 65 kD in the gel panel.

Figure 2. The detection method for the gel bands should be described in the legend.

Line 177. “Pulled on” is incorrect terminology. Rephrase.

Line 179. This is a nice demonstration.

Line 186. Please define DSBU in the text and indicate its key properties (crosslinking range, protein attachment group).

Line 189. Some explanation as to why HT2 yielded no crosslinks should be supplied. This is a surprising result given the pulldown experiments.

Figure 3. This figure is under labeled. It would be helpful to identify the termini of the crosslinks shown by the lines so that the reader can correlate these to the data in Extended Data Table 1. It is also not clear what the color coding for the lines (green, orange, black, arrows) represents. This should be explained in the legend. Finally, the black dotted lines in panel c are difficult to see.

Line 194. It might be good to reference Extended Data Figure 5c here showing the similarities between the Hsp90/Hsp70 arrangement and the proposed arrangement for GRP94 and BiP.

Line 195. Wrong PDB code.

Line 206. I'm not sure that Figure 1d supports such a categorical statement. Perhaps a rephrasing would be appropriate.

Lines 191-219. These paragraphs describe the crosslinking experiments between GRP94 and BiP, and attempt to correlate them with crosslinks that are specific to GRP94 conformations. I find it confusing that Line 219 concludes that GRP94 is in a variety of conformations yet the modeling studies say that BiP binding is only compatible with the semi-closed state. It's not clear what the crosslinking is telling us that modeling alone does not.

Line 224. Please give amino acid identities to the residue numbers.

Line 233. The rationale for including HT2 in the samples for EM is not strong, given that the SEC data in Figure 1c shows a strong BiP/GRP94 interaction.

Line 240. “Composed” is misspelled.

Line 241. How were the compositions of CI and CII determined? Is HT2 in these complexes? Is there any indication of the stoichiometry of these complexes?

Extended Data Table 1. It would be helpful to explain what the crosslink scores represent and how they were calculated.

Version 1:

Decision Letter:

20th Mar 2025

Dear Dr. Hellerschmied,

Thank you for your letter concerning your manuscript "Conformational plasticity of a BiP-GRP94 chaperone complex". We have now had a chance to discuss the points you raised in detail, and we have decided to send your paper out to review. Before we can do so, we ask for some changes to the files and documentation. Please see link below to re-submit the manuscript files.

We ask that you split the cover letter from the point-by-point response to the reviewers, and that you reiterate the reviewer concerns verbatim, in full.

Please ensure you provide relevant source data (details below).

We want to ensure that the methods and statistics reporting in our papers are of the highest quality. To that end, we ask authors to fill out a Reporting Summary that collects information on experimental design and reagents, as well as an editorial Policy Checklist, which confirms compliance with our editorial policies, including the declaration of Competing Interests.

These documents can be found by following the links below:

Reporting Summary:

Editorial Policy Checklist:

<https://www.nature.com/documents/nr-editorial-policy-checklist.pdf>

Please complete the relevant forms and return them within 48 hours. Please note that these forms are dynamic 'smart pdfs' and must, therefore, be downloaded and completed in Adobe Reader. We will then flatten them for ease of use by the reviewers. If you would like to reference the guidance text as you complete the template, please access these flattened versions at <http://www.nature.com/authors/policies/availability.html>.

Note that you are not required to revise your paper to include the information provided in the reporting summary. However, all points on the policy checklist must be addressed; please send me a new version of the manuscript with your completed checklist if needed.

Please note that all key data shown in the main figures as cropped gels or blots should be presented in uncropped form, with molecular weight markers. These data can be aggregated into a single supplementary figure item. While these data can be displayed in a relatively informal style, they must refer back to the relevant figures. These data should be submitted with the final revision, as source data, prior to acceptance, but you may want to start putting it together at this point.

Data availability: this journal strongly supports public availability of data. All data used in accepted papers should be available via a public data repository, or alternatively, as Supplementary Information. If data can only be shared on request, please explain why in your Data Availability Statement, and also in the correspondence with your editor. Please note that for some data types, deposition in a public repository is mandatory - more information on our data deposition policies and available repositories can be found below:

<https://www.nature.com/nature-research/editorial-policies/reporting-standards#availability-of-data>

Once we receive these documents and review them to ensure that all requested information is provided, we will proceed to send your paper for review. If you have questions or anticipate delays, please let me know as soon as possible.

You can use the link below to be taken directly to the site and submit your manuscript:

Link Redacted

Sincerely,

Katarzyna Ciazynska, PhD
(she/her)
Senior Editor
Nature Structural & Molecular Biology
<https://orcid.org/0000-0002-9899-2428>

Version 2:

Decision Letter:

Our ref: NSMB-A48806B

11th Apr 2025

Dear Dr. Hellerschmied,

Thank you for submitting your revised manuscript "Conformational plasticity of a BiP-GRP94 chaperone complex" (NSMB-A48806B). It has now been seen by the original referees and their comments are below. You will see that we recruited an additional reviewer #4. This was to replace the original reviewer #3 who was unavailable to assess the revision. We asked the reviewer #3 to focus on your responses to reviewer's #3 concerns. The reviewers find that the paper has improved in revision, and therefore we'll be happy in principle to publish it in Nature Structural & Molecular Biology, pending minor revisions to satisfy the referees' final requests and to comply with our editorial and formatting guidelines.

We are now performing detailed checks on your paper and will send you a checklist detailing our editorial and formatting requirements in about 2-3 weeks. Please do not upload the final materials and make any revisions until you receive this additional information from us.

To facilitate our work at this stage, it is important that we have a copy of the main text as a word file. If you could please send along a word version of this file as soon as possible, we would greatly appreciate it; please make sure to copy the NSMB account (cc'ed above).

Sincerely,

Katarzyna Ciazynska, PhD
(she/her)
Senior Editor
Nature Structural & Molecular Biology
<https://orcid.org/0000-0002-9899-2428>

Reviewer #1 (Remarks to the Author):

All my comments have been addressed adequately and the manuscript has improved substantially. There remain only a few minor typos/unclarities:

1. line 128: 'BiP shifts to the SBD open, domain-undocked conformation' -> shouldn't this be the domain-docked state?

2. line 165: 'Figure 3b shows...' -> Should be Fig. 2b

3. line 176: '...the ATP 171 lid (residues K161 and K168) to the BiPNBD lobe I' -> K213 in BiP seems to be located in the lobe IIA. Please check and clarify.

4. line 216: 'Figure 3d, ...' -> Should be Fig. 2d

Reviewer #2 (Remarks to the Author):

The revised manuscript by Brenner et al. describes the use of crosslinking mass spectrometry and cryoEM to investigate the binding of GRP94, an Hsp90 chaperone, to BiP, an Hsp70 chaperone. The authors have added significant data to the manuscript, which strengthens the overall conclusions and the impact of the research.

The revised manuscript addresses all concerns from this reviewer and seems to address the points raised by the other reviewers as well.

Reviewer #4 (Remarks to the Author):

This study combines XL-MS and negative staining to characterize BiP-GRP94 complexes. They followed up on their observations using an unfolded substrate to understand client loading and processing. This work is highly novel and provides mechanistic insight into the co-operation between Hsp70 and Hsp90 family proteins. The authors have gone to great lengths to address the substantial comments of the original three reviewers and in doing so have produced a greatly improved version of their manuscript. I congratulate all the authors on this revision, and I appreciate the time and effort that went into producing this wonderful study! Andy Truman

We greatly appreciate the reviewers' interest in our findings and thank them for their constructive feedback. Addressing their key concerns has enhanced the manuscript and further solidified our conclusions.

Changes and additions in the revised manuscript are highlighted in **yellow**. Key new figures are additionally included in the point-by-point response below.

Reviewers' Comments:

Reviewer #1:

Remarks to the Author:

Brenner et al provide a biochemical and structural analysis of the interaction between the chaperones BiP and GRP94, which are critical for the correct folding of proteins in the secretory pathway. These two proteins cooperate but the structural understanding of these chaperoning machines and their interplay has been limited by their high conformational flexibility.

The key findings of the manuscript are (i) that BiP and GRP94 may form a complex very similar to the loading complex previously described for the cytosolic paralogs Hsp70 and Hsp90, (ii) that to get there, BiP and GRP94 don't seem to need the assistance of cochaperones like Hop, which is essential for the assembly of Hsp70-Hsp90 complexes, and (iii) the identification and structural description of a pre-loading complex that may precede the observed loading complexes. These observations are important and will be influential to the chaperone field. However, the manuscript could improve substantially by solidifying the relevance of the observations as well as by some minor amendments.

Major issues:

1. Did the author also test a GRP94 $\Delta 72$, where the purification tag was removed (other than Fig. 4f)? This is important as the tag might have been crucial for stabilized the observed complex. But its stabilization might also be non-physiological.

R1.1 We thank the reviewer for raising this important point. In the revised manuscript we provide XL-MS data on the GRP94 full-length protein and the GRP94 $\Delta 72$ lacking the purification tag (GRP94 $\Delta 72^{\text{noPT}}$), both of which cannot show an artificial interaction with a purification tag (see also **R1.4c**). These new XL-MS data show that the purification tag stabilizes the complex, but the BiP_{NBD}-GRP94 interfaces are not affected by this stabilization i.e. the interaction interfaces I and II, which define the BiP-GRP94 pre-loading and loading complexes, are preserved in the absence of the GRP94 $\Delta 72$ purification tag. Full-length GRP94 also interacts with BiP in the SEC experiments shown in Figure 1. When in complex with the full-length GRP94 protein, the BiP_{SBD} extends towards the GRP94_{CTD} (in the absence of the artifactual interaction with the purification tag), consistent with a substrate-transfer from BiP to GRP94. Additionally, we performed experiments studying the effect of ATP analogues and PU-WS13 on complex formation with GRP94 $\Delta 72^{\text{noPT}}$ (see below **R1.3** and **R1.7**).

2. Why did the authors use the BiP_{NBD} construct in the chemical (glutaraldehyde) crosslinking when it didn't bind on its own. The authors should repeat this with the BiP-GRP94 $\Delta 72$ constructs used later for structural studies

R1.2 We added chemical crosslinking data of the constructs used for structural studies (BiP and GRP94 $\Delta 72$) to Extended Data Figure 1.

Extended Data Figure 1. (b) Analytical SEC and SDS-PAGE analysis of a BiP-GRP94 $\Delta 72$ sample crosslinked with DSBU. Protein bands on SDS-PAGE gels were visualized by Coomassie staining.

Using the BiP_{NBD} in chemical crosslinking experiments enabled us to study the effect of AMP-PNP-bound GRP94 in BiP-GRP94 complex formation – see also **R1.6**. We now provide our rationale for using the BiP_{NBD} in the crosslinking experiments involving ATP analogues in the revised manuscript (lines 125-131). The BiP_{NBD} binds on its own, however the complex does not withstand the conditions of size exclusion chromatography (SEC) (e.g. dilution of the sample). While it does not add to the conclusions of the manuscript, we would like to keep the negative SEC data for transparency (Extended Data Figure 2c).

3. The authors should test the effect of an GRP94 inhibitor on complex formation (e.g. the chemical (glutaraldehyde) crosslinking) with would be expected block the closed conformation

R1.3 We thank the reviewer for this suggestion. We tested the effect of the GRP94 inhibitor PU-WS13 on complex formation between BiP and GRP94 in chemical crosslinking experiments. In these experiments we used the GRP94 $\Delta 72^{\text{noPT}}$ construct, where the purification tag was removed (GRP94 $\Delta 72^{\text{noPT}}$) and the GRP94 FL protein. These additional data show that the GRP94 inhibitor, which is expected to prevent full closure of the GRP94 N-domains, stabilizes a BiP-GRP94 complex (Figure 1e). This effect aligns with the closed GRP94 structure counteracting complex formation with BiP (Figure 1d).

Figure 1. (e) Left panel: Chemical crosslinking of GRP94 $\Delta 72^{\text{noPT}}$ and BiP_{NBD} in the presence of PU-WS13. **Right panel:** Quantification of the three different species indicated on the gel. (n = 3, mean + sd). Protein bands on SDS-PAGE gels were visualized by Coomassie staining.

4. The XL-MS experiments need control experiments and a much better representation/explanation:

a) Fig. 3a: Use different colors (for the different types of inter-BiP-GRP94 crosslinks) to indicate, which crosslinks correspond to interface I, interface II, and to the purification tag. Why do the authors observe so much more crosslinks for the GRP94-NTD/BiP-SBD and GRP94-PT/BiP-SBD compared to the interfaces I and II, which are at the heart of this manuscript.

R1.4a We thank the reviewer for this suggestion. We introduced a color-based categorization of the crosslinks to distinguish the different BiP-GRP94 crosslinks (Figure 2 and Extended Data Figure 3).

Specifically, residues S2 and K16 in the GRP94 Δ 72 purification tag appear very reactive and prone to crosslinking. Given the expected flexibility of the tag, it crosslinks to various residues within the BiP_{SBD}, as shown in Figure 3a. In general, the purification tag appears to be very amenable to DSBU-based crosslinking, as it also participates in non-specific intramolecular crosslinks to GRP94 (see Extended Data Table 2). Moreover, in our new BiP-GRP94 FL XL-MS dataset, it appears that crosslinks to the BiP_{SBD} are also more frequent (while they are placing the BiP_{SBD} in close proximity of the GRP94_{MD/CTD}). The more prominent crosslinking to the GRP94 Δ 72 purification tag may therefore also reflect a property of the BiP_{SBD}.

Notably, crosslinks outlining interface I and II are identified in all our XL-MS datasets (Extended Data Tables 1-4), even in the absence of the artifactual BiP-purification tag interaction.

b) The observed crosslinks between BiP and GRP94 Δ 72 suggest that the two chaperones adopt a similar conformation as their cytosolic paralogs, Hsp70 and Hsp90, in the previously described loading complex. Please amend ED Tab1. to include a comment whether the particular crosslinks are compatible (or not) with the homology modeled based on PDB-ID 7kw7. Please also include, which distance cutoff will be used for this assessment.

R1.4b We have revised the Extended Data Tables to include distance measurements and according comments on the compatibility with the loading complex conformation (see Extended Data Tables 1-4).

c) Please repeat this analysis with wildtype GRP94 and BiP. I consider this much more relevant than the experiment with/without HT2. It helps to support the proposed structures in the absence of the (potentially artificial) stabilization by the tag.

R1.4c We fully agree with the reviewer and have performed the experiment and analysis with the GRP94 full-length (FL) and BiP proteins accordingly. These new data are presented in Figure 2 and Extended Data Figures 3 and 4. Consistent with our previous data obtained for the BiP-GRP94 Δ 72 complex, the XL-MS data of BiP-GRP94 FL show the formation of interface I and II between BiP_{NBD} and GRP94_{NTD} and GRP94_{MD} (Figure 2a and 2c). The position of the BiP_{SBD} differs between the BiP-GRP94 FL and BiP-GRP94 Δ 72 complexes. As reported in the original manuscript, when in complex with GRP94 Δ 72, the BiP_{SBD} points towards the GRP94_{NTD} and the N-terminal GRP94 Δ 72 purification tag. Interestingly, our new data show that when in complex with GRP94 FL, the BiP_{SBD} is positioned in proximity of the GRP94_{MD/CTD}, a conformation compatible with substrate loading onto GRP94, as described for the cytosolic loading complex.

Figure 2. Crosslinking mass spectrometry outlines the topology of a BiP-GRP94 complex. (a) Map of crosslinks identified in the BiP-GRP94 FL (left panel) and BiP-GRP94 Δ72 (right panel) samples. (b) Homology model of BiP-GRP94 on the basis of the cytosolic loading complex (PDB ID: 7KW7). The ATP lid is indicated in red. Residues indicated as the predicted GRP94 substrate binding region (dark green) include the amphipathic helix N657-Q668 as identified in the cytosolic Hsp90 paralog and additional residues F398-Y401, K428, I497, E498, and Y575 as identified in Huck et al.³² (c) Crosslinks outlining interface I, interface II, and the ATP lid-BiP_{NBD} interactions from the BiP-GRP94 FL XL-MS dataset mapped onto the homology model shown in (b). BiP_{SBD} omitted for clarity. (d) Crosslinks from the BiP_{SBD} to GRP94 mapped onto the homology model shown in (b). BiP_{NBD} omitted for clarity. (e) Summary of XL-MS results. NTD: N-terminal domain, MD: middle domain, CTD: C-terminal domain, NBD: nucleotide-binding domain, SBD: substrate binding domain, PT: purification tag of GRP94, CL: charged linker, L: linker, H: His₆-tag, S: Strep-tag. Crosslinks to GRP94 protomer A are shown, which represent the closest distances for all residues shown in panel (c) and (d) except for K733.

Minor issues:

5. I hardly see a difference in BiP co-elution percentage between GRP94wt vs D149N vs E103A (Fig. 1b & ED1b). If there was a minor increase in BiP co-eluting with GRP94 D149N, why did the author not use this mutant later on (I assume the following experiments were performed with GRP94 containing a wildtype NTD).

R1.5 We initially used the GRP94 ATP binding and hydrolysis mutants to study the effect of the GRP94 open and closed conformation on BiP-GRP94 complex formation. Indeed, the mutant proteins did not allow us to draw any strong conclusions regarding this aspect. Upon close inspection of all our data on the interaction of BiP with the corresponding GRP94 mutant proteins, we have rephrased our interpretation to better reflect the data shown in Figure 1 and Extended Data Figure 1: *Compared to GRP94 wild-type protein we did not observe major differences in BiP co-eluting with GRP94 D149N or E103A (Extended Data Figure 1a).*

To conclusively study the effect of the GRP94 closed state and a condition expected to prevent closure, we performed chemical crosslinking with the BiP_{NBD} in the presence of ATP analogues and the GRP94 inhibitor PU-WS13. An according quantification of this new data is shown in Figure 1de.

Yes, all following experiments were performed with a GRP94 containing the wild-type NTD sequence.

6. To further assess the effect of the GRP94 nucleotide-bound state on complex formation we used a BiP_{NBD} construct, which irrespective of its nucleotide state, retains GRP94 binding. Please explain why the authors used this construct and GRP94 binding is irrespective of its nucleotide state here

R1.6 The full-length BiP protein can reside in two extreme states – the fully extended (ADP) and lid open (ATP) state. When bound to ATP the BiP_{SBD} blocks the interaction surface on the BiP_{NBD}, which is required for GRP94 interaction. Upon deletion of the BiP_{SBD}, the BiP_{NBD} can bind to GRP94 when bound to ADP, ATP, or ATP analogues like AMP-PNP. We therefore chose the BiP_{NBD} truncation construct to assess the effect of ATP analogues (binding to GRP94) on GRP94-BiP complex formation. We have added an according clarification to the revised manuscript (lines 125-131).

7. with an even more pronounced band corresponding to a 133 GRP94 Δ 72 dimer with two BiP_{NBD} molecules bound. I find the evidence for two BiP_{NBD} molecules not that strong; the bands are rather weak compared to the crosslinked GRP94 dimer alone

R1.7 We have quantified the relative abundance of the three species observed on the SDS-PAGE gel – (1) GRP94 dimer by itself and (2) bound to one or (3) two BiP_{NBD}s – to show differences in their distribution under different conditions (Figure 1d and e). All of the GRP94 protein shifts to the dimer species in the crosslinking experiment as GRP94 is constitutively dimerized via its C-terminal domains. We presume that the complexes with the BiP_{NBD}s are less abundant on the gel, as they are less stable than the constitutive GRP94 dimer.

Figure 1. (d) *Left panel:* Chemical crosslinking of GRP94 $\Delta 72^{\text{noPT}}$ and BiP_{NBD} in the presence of ADP or AMP-PNP. *Right panel:* Quantification of the three different species indicated on the gel. (n = 3, mean + sd)

8. The authors work on the HT2-tag system is beautiful. It is unfortunate that the HT2 substrate could not be detected in the XL-MS or the cryo-EM structures. The lack of well defined and tractable authentic substrates for BiP/GRP94 is a major limitation for this branch of the chaperone field.

R1.8 Thank you. We think that the reason for this unfortunate result is two-fold.

HT2 contains few lysines (7 lysines, 2.3% of its sequence) likely preventing efficient crosslinking with glutaraldehyde and DSBU in the GraFix and XL-MS experiments, respectively. To increase the number of lysine residues and the size of the substrate protein, we turned to a firefly luciferase-HT2 (LUC-HT2) fusion protein. We previously demonstrated that purified LUC-HT2 is destabilized upon hydrophobic tagging (Figure S1 in Serebrenik et al. doi: [10.1091/mbc.E17-11-0693](https://doi.org/10.1091/mbc.E17-11-0693)). We screened for conditions that would allow soluble BiP-GRP94-LUC-HT2 complex formation and crosslinking. After extensive optimization, we were able to identify a minimal number of crosslinks from LUC-HT2 to BiP/GRP94 (**Figure Rev 1**). Our results support the hypothesis that the HT2 sequence precludes crosslink identification since all crosslinks found between BiP/GRP94 and LUC-HT2 involve the luciferase part of the construct and very few crosslinks are found for HT2 overall (**Figure Rev 1, top panel**). The minimal crosslinks to luciferase did not allow us to place the substrate protein in the BiP-GRP94 complex. We have therefore not included these data in the manuscript.

HT2 may additionally be shielded by the BiP and GRP94 substrate binding regions, which may hinder the penetration of the cross-linker near the binding sites. A parallel approach using a heterobifunctional crosslinker (Sulfo-LC-SDA) that reacts with primary amines and carries a photo-crosslinking moiety was not successful in crosslinking complexes between BiP, GRP94, and HT2. While this result is not direct evidence, it also supports this hypothesis.

We agree with the reviewer that BiP/GRP94 substrates are strongly required to better understand these chaperones. We think that HT2 fusion proteins offer a promising enhancement to the system and they will be further explored in future research. Our conclusions regarding benefits and limitations of using the HT2-hydrophobic tagging system in studying chaperone-substrate complexes are summarized in the discussion section (lines 485-494).

Figure Rev1: Crosslinking results of LUC-HT2 with chaperones BiP (top panel) and BiP/GRP94 (bottom panel). 8 μ M Luciferase-HaloTag2 (LUC-HT2) was reacted with 12 μ M HyT36 and incubated with 4.0 μ M BiP or 4.0 μ M BiP and 4.0 μ M GRP94 Δ 72^{noPT} for 60 min at 30 °C (to induce misfolding and to avoid aggregation of the LUC-HT2 construct, which was observed at 37° C). DSBU was added to a final concentration of 0.5 mM for the last 30 min before quenching with 20 mM Tris pH 8.0. High molecular weight bands were cut from SDS-PAGE gels and subjected to MS analysis as described in the manuscript (samples were also digested in solution and submitted to MS, but did not yield any different results). Data were analyzed with Xisearch and the figure generated with Xiview. Crosslinks with a Xisearch score >15 are shown. PT: purification tag of GRP94 (which is cleaved off for this experiment), NTD: N-terminal domain, MD: middle domain, CTD: C-terminal domain, NBD: nucleotide-binding domain, SBD: substrate binding domain, CL: charged linker, L: linker.

9. ...we performed Gradient Fixation (GraFix), which resulted in the separation of three major complex species. I don't see a real separation of the complex species (by the GraFix procedure)

R1.9 We thank the reviewer for pointing this out. We agree and have changed the text accordingly to report the *formation and enrichment* of the complex species (lines 285-287).

10. ... Upon removal of the GRP94 purification tag, we observed a reduction in GRP94 protein bound to BiP by ~65%, suggesting that the tag contributes to the stability of the chaperone complex.

Could this represent primarily a complex with only one BiP bound compared to a general destabilization of the complex?

R1.10 This is an interesting hypothesis. From our current data, unfortunately we cannot definitively answer this question.

11. We presume that interactions between substrate protein and GRP94 drive the formation of the loading complex, in addition to interactions between the GRP94NTDs and BiPNBDs, further facilitating substrate transfer. Please discuss how these processes might be coupled

R1.11 We added a more detailed discussion of this aspect (lines 499-508).

12. ...our structural and biochemical data provide insight into the BiP-GRP94 pre-loading and loading complex conformations, two successive states along the GRP94 chaperone cycle... This is indeed tempting to postulate - do the author have any data for supporting a transformation of pre-loading to loading complex? Perhaps this should be phrase a bit more cautiously.

R1.12 We agree and have removed the term "successive" to avoid overinterpretation of the data (line 427). The possible sequence of events is now solely discussed as a hypothesis in the final model.

13. Please address in the discussion (and in Fig. 6) - where possible - the role of ADP/ATP states and ATP hydrolysis/nucleotide release in the chaperoning cycle

R1.13 We thank the reviewer for this suggestions. We have addressed the ADP/ATP states of GRP94 and BiP in the discussion (lines 434-436, 447-456) and in the figure legend of Figure 6.

Reviewer #2:

Remarks to the Author:

The manuscript by Brenner et al. describes a combination of crosslinking mass spectrometry (XL-MS) and negative-stain electron microscopy (EM) for the analysis of GRP94-BiP chaperone complexes. By using hydrophobic tagging to include an unfolded substrate in the reconstituted system, the authors were able to isolate the complex in the pre-loading and loading complex states and both biochemically and structurally characterize these forms. They identify critical interaction sites between GRP94 and BiP and compare these interactions with those for HSP90 and HSP70. Overall, the XL-MS and EM results are in good agreement and provide unique insights into the conformations.

Major Comments:

1. The authors assume that intramolecular crosslinks in GRP94 are generally interprotomer rather than intraprotomer, as assessed from the indicated crosslinks in Figure 3. How did the authors determine which of the crosslinks shown in Figure 3 are intraprotomer and which were interprotomer? Although these come from high molecular weight bands, can the authors necessarily exclude that the intraprotomer crosslinks come from two different molecules rather than one? For example, were monomer crosslink bands also analyzed to see what intramolecular crosslinks form there to support that these are interprotomer? If not, then the results need to be amended to clarify that the assignment of intraprotomer or interprotomer is hypothetical and that an alternate assignment is possible.

R2.1 In the revised manuscript, we provide a more detailed description of our analysis of the crosslinking data (lines: 178-204). We have also added distance measurements (inter- and intra-protomer) for all identified GRP94 crosslinks in the open and semi-closed GRP94 dimer. The interpretation of these measurements is also added as a column with comments to the Extended Data Tables 1-4. We considered GRP94 crosslinks as intra- and interprotomer crosslinks in our analysis.

2. Were any other crosslinkers used to attempt to get some crosslinks with HT2? Do the authors have any thoughts on whether the lack of HT2 crosslinks is due to technical challenges of the experiment itself (no good crosslinking results in HT2 proximal to where it might bind, for example) or instability in the complex that prevents such crosslinks from readily forming and being detected?

R2.2 We believe that the low number of lysine residues in the HT2 sequence is a major limitation in our XL-MS experiments. However, we were also unsuccessful in establishing conditions to stabilize a BiP-GRP94-HT2 complex using a heterobifunctional crosslinker attaching to primary amines and carrying a photo-reactive group (Sulfo-LC-SDA). This may be related to shielding of HT2 by the BiP/GRP94 substrate binding sites. Please refer to **R1.7** for a detailed response to this matter.

3. Tables of the intramolecular crosslinks in the Extended Data should be included.

R2.3 All Extended Data Tables containing crosslinking data now also include the analyses of intra-molecular crosslinks.

4. Are the crosslinks of BiP with the GRP94d72 purification tag consistent with the stabilization effect observed?

R2.4 In the revised manuscript, we added a XL-MS dataset of a BiP-GRP94 FL sample. We can now directly compare the crosslinking data (Figure 2a) and pull-down data (Figure 4f) between the BiP-GRP94 FL and BiP-GRP94 Δ 72 construct which contains the purification

tag interacting with BiP. Differences in the crosslinking pattern are only observed in the BiP_{SBD} region. We therefore think that the crosslinks are consistent with the stabilization effect. As also shown in Figure 4e, crosslinks agree with the purification tag being in close proximity to the BiP_{SBD}.

5. Were the XL-MS experiments repeated on separately-prepared samples, or do these represent results from a single set of samples?

R2.5 XL-MS experiments were repeated on three separately-prepared BiP-GRP94 Δ 72 samples and two separately-prepared BiP-GRP94 FL samples. This is now reported in the Extended Table legends. The XL-MS results consistently outlined the same interaction interfaces and protein conformations. Samples containing PU-WS13 (Extended Data Tables 3 and 4) were measured once.

Minor Comments:

6. In Extended Data Table 1, it seems that the scores from Xisearch are color coded, but that is not described.

R2.6 In the course of preparing the revised manuscript we decided to omit the color code for the score for clarity of the Extended Data Tables.

7. Page 6, lines 214 and 217 talking about the crosslinks between the GRP94MD GRP94NTD refer to Fig 3b, should be 3c?

R2.7 Thank you for noting. In the course of preparing the revised manuscript the figure numbering has changed and we have updated the figure references accordingly.

Reviewer #3:

Remarks to the Author:

This manuscript describes an in depth characterization of the interaction between the ER chaperones BiP and GRP94. The authors use size exclusion chromatography, chemical crosslinking combined with mass spectrometry, modeling studies, and negative stain electron microscopy to show that BiP has multiple interaction surfaces on GRP94. The authors also investigate the interaction between a model unfolded substrate, HT2, and the BiP/GRP94 complex and show that they can isolate a complex containing all three of the proteins.

Overall this is a technical tour de force. The formation of the ternary complex with HT2 is a terrific achievement. On the other hand, I have several reservations about the paper that should be considered before proceeding further:

First, while the in vitro complex with the model unfolded substrate HT2 is a novel result, it is not clear whether this reflects a biologically relevant state or activity for GRP94. The role of GRP94 in protein homeostasis and misfolding in the ER is, unfortunately, still very unclear. The authors cite references 46 and 47 (line 142) as evidence that GRP94 interacts with HT2 in HEK cells, but I was not able to find a convincing description of this specific interaction in either of these reports, which appear to focus mainly on the interaction with BiP. (GRP94 is not mentioned in either of these references, and the implication is that the KDEL proteins also identified include GRP94.) If GRP94 is not involved in misfolded protein correction or clearance, then the employment of the HT2 substrate is a clever approach to the substrate problem but ultimately artificial. As such, the corollary is that these studies may not do much to illuminate the role of GRP94 in client maturation.

R3.1 To address this point, we included a pull-down experiment from HEK293 cells, which demonstrates an interaction of HT2 and GRP94. This chaperone-substrate interaction hinges on the folding state of HT2 (Extended Data Figure 5, lines 229-237).

Second, despite demonstrating that HT2 is captured by BiP/GRP94 (Figure 2e, panel 2 is terrific), the authors are unable to further characterize this interaction. HT2 does not show up in any of their crosslinking studies, and it is not visible in the electron microscopy reconstructions. This is a quite surprising given the fact that the complex appears to be somewhat stable.

R3.2 We agree with the reviewer and we have performed additional experiments to clarify this point. Please refer to **R1.8** for a response.

Third, while technically laudable, and clearly the product of a great deal of effort, the overall result of this study does not, unfortunately, add much to our understanding beyond what could have been achieved by homology modeling with the referenced Hsp90/Hsp70/GRP complexes. It does not appear that any new interaction surfaces are identified, the location of the HT2 model substrate is not revealed, and there is also an artificial BiP/GRP94-His tag interaction that may affect the positioning of the BiP SBD in their models.

R3.3 While homology modeling of BiP-GRP94 on the basis of the Hsp70/Hsp90/Hop/GRP complex helped support the interpretation of our data, our experimental results provide crucial insights that cannot be captured by homology modeling alone. 1) In addition to the Hsp70/90 loading complex conformation, our experimental data identify the BiP-GRP94 pre-loading complex conformation. 2) Our XL-MS data show that the BiP-GRP94 pre-loading and loading complex conformations exist in solution and form in the absence of any co-chaperone or substrate protein (These data were consistently obtained in the presence and absence of the artificial BiP_{SBD}-purification tag interaction). 3) While artificial, the BiP_{SBD}-purification tag interaction shows that the BiP_{SBD} is highly flexible in complex with GRP94,

likely reflecting the diverse range of substrate proteins engaged by this chaperone system. Additional XL-MS data in the revised manuscript show a preference of the BiP_{SBD} in extending towards the predicted substrate binding region of GRP94, even in the absence of substrate protein (in the absence of BiP_{SBD}-purification tag interaction). 4) Our chemical crosslinking data demonstrate that AMP-PNP-bound GRP94 does not interact with BiP (specifically, with the BiP_{NBD}), while PU-WS13, an inhibitor that prevents closure of the GRP94_{NTDS}, favors a BiP-GRP94 complex.

Detailed comments are listed below:

Line 107. In Figure 1b the BiP SEC line is hard to see. Perhaps a thicker line or different color could be used.

R3.4 For better visibility, we changed the color and thickness of the BiP SEC lines in all figures.

Line 117, 118. It would be helpful if the term „minor” could be quantified by giving a percent. These proportions are hard to determine in the Figure. Same as line 117 comment for the term „reduced”

R3.5 We thank the reviewer for this comment. We have adjusted our description of the SEC data. Please refer to **R1.5** for details.

Line 131. I'm afraid that I wasn't able to see the „strongly reduced” complex the authors are referring to. Perhaps quantitation of the relevant supershifted complex bands would be helpful here.

R3.6 We have quantified the relative abundance of complexes containing GRP94 dimer crosslinked to one or two BiP_{NBD} domains in the experiments shown in Figure 1de accordingly. Please also refer to **R1.7**.

Figure 1. The authors should indicate the method of detection used α C staining or blot?

R3.7 Protein bands on SDS-PAGE gels were visualized by Coomassie staining, which is now reported in the figure legends.

Line 135. „Protocol” might be replaced with something like „characterized the interaction.”

R3.8 Additional data were included into this paragraph and the final sentence was rephrased accordingly (lines 149-152).

Line 142. References 46 and 47 do not seem to show GRP94 involvement with HT2 in cells.

R3.9 We apologize for this oversight. Indeed, in reference 46 (54 in the revised manuscript) interactors of an HT2 fusion protein expressed in HeLa cells were detected with an antibody recognizing the C-terminal KDEL motif present in ER chaperones, as noted by the reviewer. Solely BiP was also visualized with a protein-specific antibody as an HT2 interactor in ref 46 and 47 (54 and 55 in the revised manuscript). We isolated B4GT-HA-EGFP-HT2, an HT2 fusion protein localized to the Golgi apparatus and the ER. We demonstrate an interaction between B4GT-HA-EGFP-HT2 and the chaperones BiP and GRP94. This interaction hinges on the folding state of HT2. The HT2 stabilizer HALTS1 reduces the isolated amounts of BiP and GRP94, while the destabilizing hydrophobic tag, HyT36 increases them. These data

establish misfolded HT2 as a substrate for BiP and GRP94 in cells, as now reported in Extended Data Figure 5 and lines 229-237.

Extended Data Figure 5. EGFP-trap pull-down of B4GT-HA-EGFP-HT2 from HEK293 cells. Western blot analysis of input and elution fractions probing for the bait protein, BiP, and GRP94.

Line 175. In Figure 2e it would be helpful to identify the band at 65 kD in the gel panel.

R3.10 The band at 65 kDa in the input fractions represents BSA, which was added to the interaction buffer for pull-down studies.

Figure 2. The detection method for the gel bands should be described in the legend.

R3.11 Protein bands on SDS-PAGE gels were visualized by Coomassie staining, which is now reported in the figure legends.

Line 177. „Pulled on“ is incorrect terminology. Rephrase.

R3.12 We rephrased the description of the sequential pull-down experiment accordingly (lines 270-276).

Line 179. This is a nice demonstration.

R3.13 Thank you.

Line 186. Please define DSBU in the text and indicate its key properties (crosslinking range, protein attachment group).

R3.14 A definition of DSBU and its key properties was added to the text (lines 157-159).

Line 189. Some explanation as to why HT2 yielded no crosslinks should be supplied. This is a surprising result given the pulldown experiments.

R3.15 We agree with the reviewer and have investigated the applicability of HT2 in crosslinking experiments. Please refer to **R1.8** for a detailed explanation. Our conclusions are summarized in the discussion section (lines 485-494).

Figure 3. This figure is under labeled. It would be helpful to identify the termini of the crosslinks shown by the lines so that the reader can correlate these to the data in Extended Data Table 1. It is also not clear what the color coding for the lines (green, orange, black,

arrows) represents. This should be explained in the legend. Finally, the black dotted lines in panel c are difficult to see.

R3.16 As also noted by reviewers 1 and 2, we have improved the representation of the crosslinking data in the revised manuscript by including a color-code for BiP-GRP94 crosslinks and adding residue numbers to figures displaying these crosslinks. Crosslinks mapped onto the open or semi-closed GRP94 conformation (panel 3c in the original manuscript) are now represented in Extended Data Figure 4 with a more detailed description in the text (lines: 178-222). See also our response to **R1.4**.

a GRP94 - open

GRP94 - semiclosed

b

Extended Data Figure 4. Crosslinking mass spectrometry outlines the conformations of BiP and GRP94. (a) Intra-GRP94 crosslinks mapped onto the open (PDB ID: 2O1V) or semi-closed (homology model based on PDB ID: 7KW7) GRP94 dimer. For simplicity, only MD and NTD are shown. Crosslinks

from the MD region in protomer B - K455 (B), K458 (B), K447 (B) to the NTD region within the same protomer and to protomer A are shown - K114 (A/B), K161 (A/B), K95 (A/B), K97 (A/B). Crosslinks shown in purple support the open conformation, crosslinks shown in orange support the semi-closed conformation. (b) Intra-BiP_{SBD} crosslinks between the SBD α and SBD β (identified in the BiP-GRP94 FL XL-MS dataset, Extended Data Table 1) mapped onto the closed (left panel, PDB ID: 5E85) and open (right panel, PDB ID: 5E84) BiP_{SBD} structure. The BiP_{SBD} domains are aligned based on the SBD β . All crosslinks indicate that the BiP_{SBD} is in the closed conformation.

Line 194. It might be good to reference Extended Data Figure 5c here showing the similarities between the Hsp90/Hsp70 arrangement and the proposed arrangement for GRP94 and BiP.

R3.17 We introduced a reference to the corresponding Figure, which is Extended Data Figure 8 in the revised manuscript.

Line 195. Wrong PDB code.

R3.18 Thank you for noting. The PDB code was corrected.

Line 206. I'm not sure that Figure 1d supports such a categorical statement. Perhaps a rephrasing would be appropriate.

R3.19 We removed the word 'strongly' to reflect the data shown in our new Figure 1d, which now also includes a quantification.

Lines 191-219. These paragraphs describe the crosslinking experiments between GRP94 and BiP, and attempt to correlate them with crosslinks that are specific to GRP94 conformations. I find it confusing that Line 219 concludes that GRP94 is in a variety of conformations yet the modeling studies say that BiP binding is only compatible with the semi-closed state. It's not clear what the crosslinking is telling us that modeling alone does not.

R3.20 In the revised manuscript we have re-written this paragraph to provide a clearer description of the crosslinking results and to better explain the conclusions drawn from them (lines: 178-222).

Modelling suggests that BiP binding to GRP94 is compatible with the open and the semi-closed GRP94 state. The crosslinking data show that complexes between BiP with the open and semi-closed GRP94 exist in solution. Accordingly, the crosslinking data experimentally support the formation of the pre-loading complex described in this manuscript, where GRP94 is in the open conformation.

We also obtained a new dataset from a BiP-GRP94 FL sample. These new crosslinking data show that even in the absence of substrate protein the BiP_{SBD} extends towards the GRP94_{MD/CTD} where the predicted substrate binding region of GRP94 is located. Moreover, new crosslinking data obtained from samples with PU-WS13 suggest that the GRP94 ATPase inhibitor stabilizes a GRP94 conformation that facilitates the formation of interface I and II in the BiP-GRP94 complex.

Line 224. Please give amino acid identities to the residue numbers.

R3.21 We added the amino acid identities.

Line 233. The rationale for including HT2 in the samples for EM is not strong, given that the SEC data in Figure 1c shows a strong BiP/GRP94 interaction.

R3.22 Indeed the data presented in Figure 1 show a strong interaction between the two chaperones. Nevertheless, at the time of sample preparation for EM, we wanted to include HT2 to further stabilize BiP in the SBD-closed conformation, which is required to bind to

GRP94. This may not have been strictly necessary, as we now know about the additional interaction between the BiP_{SBD} and the purification tag.

Line 240. „Composed“ is misspelled.

R3.23 Thank you for noting. We corrected the spelling.

Line 241. How were the compositions of CI and CII determined? Is HT2 in these complexes? Is there any indication of the stoichiometry of these complexes?

R3.24 In Western blot analysis of the GraFix fraction used for EM, we detected a limited amount of HT2. This analysis was added as Extended Data Figure 6b. HT2 is present at substoichiometric amounts, which is indicated by the SDS-PAGE gel showing samples, which were not crosslinked with glutaraldehyde (Extended Data Figure 6a). Complexes of GRP94 with one or two BiP molecules were the most abundant structures in datasets obtained from grids prepared from fraction 12 and we therefore assigned the complex I and II composition accordingly.

Extended Data Figure 6. (b) Western blot analysis of input and gradient fractions 12, 13, and 14 containing glutaraldehyde. The use of different gels accounts for the different banding pattern compared to (a). The asterisk denotes potentially aggregated HT2 protein.

Extended Data Table 1. It would be helpful to explain what the crosslink scores represent and how they were calculated.

R3.25 Crosslink scores were calculated in Xisearch (Mendez et al. 2019 doi: 10.15252/msb.20198994). A brief description of the scoring function is now reported in the methods section titled *XL-MS data analysis* (lines 726-731).

We would like to thank all the reviewers for carefully assessing our revised manuscript and for their final comments. We really appreciate their input, which greatly improved the manuscript as well as their encouraging words. A point-by-point response to the remaining issues is enclosed.

Reviewer #1 (Remarks to the Author):

All my comments have been addressed adequately and the manuscript has improved substantially.

Thank you.

There remain only a few minor typos/unclarities:

1. line 128: 'BiP shifts to the SBD open, domain-undocked conformation' -> shouldn't this be the domain-docked state?

Thank you for noting, we have changed the term accordingly referring now to the domain-docked state.

2. line 165: 'Figure 3b shows...' -> Should be Fig. 2b

We have changed the Figure reference to Fig. 2b.

3. line 176: '...the ATP 171 lid (residues K161 and K168) to the BiPNBD lobe I' -> K213 in BiP seems to be located in the lobe IIA. Please check and clarify.

Thank you for pointing this out. We have rephrased the text to reflect that in the first sentence we specifically refer to the formation of interface II (which involves lobe I) and in the following sentence additional crosslinks (to residue K213, located in lobe IIA) are discussed:

'... and the ATP lid (residues K161 and K168) to the BiP_{NBD} lobe I. *Additional* crosslinks between BiP residue K213 and GRP94 residues K161 and K168 suggest that the GRP94 ATP lid may be slightly closer to the BiP_{NBD} than shown in the homology model...'

4. line 216: 'Figure 3d, ...' -> Should be Fig. 2d

We have changed the Figure reference to Fig. 2d.

Reviewer #2 (Remarks to the Author):

The revised manuscript by Brenner et al. describes the use of crosslinking mass spectrometry and cryoEM to investigate the binding of GRP94, an Hsp90 chaperone, to BiP, an Hsp70 chaperone. The authors have added significant data to the manuscript, which strengthens the overall conclusions and the impact of the research.

The revised manuscript addresses all concerns from this reviewer and seems to address the points raised by the other reviewers as well.

Thank you.

Reviewer #4 (Remarks to the Author):

This study combines XL-MS and negative staining to characterize BiP-GRP94 complexes. They followed up on their observations using an unfolded substrate to understand client loading and processing. This work is highly novel and provides mechanistic insight into the co-operation between Hsp70 and Hsp90 family proteins. The authors have gone to great lengths to address the substantial comments of the original three reviewers and in doing so have produced a greatly improved version of their manuscript. I congratulate all the authors on this revision, and I appreciate the time and effort that went into producing this wonderful study! Andy Truman

Thank you.